# A functional overlap between actively transcribed genes and chromatin insulator elements

Lucy J Cornell[1,5], Caroline L Harrold[1,5], Susannah Holliman[1], Felice Tsang[1], Matthew E Gosden[1], Lars L P Hanssen [1], Rosa Stolper[1], Damien J Downes [1], Daniel Biggs[2], Chris Preece[2], Samy Alghadban[2], Jacqueline A Sharpe[1], Benjamin Davies [3], Jacqueline A Sloane-Stanley[1], Jim R Hughes [1,4 ✉], Douglas R Higgs [1 ✉] & Mira T Kassouf [1 ✉]

## Abstract

**The mammalian genome is organised into large topologically associating domains (TADs) and smaller sub-TADs or enhancer-promoter loops, which may contribute to the regulation of gene expression. These dynamic structures arise, at least partly, via cohesin-mediated loop extrusion delimited by insulator elements. By studying the structure and function of the alpha-globin locus during erythroid differentiation, we have previously shown that the juxtaposition of the enhancers and promoters during this process partly depends on cohesin-mediated loop extrusion, which appears to be delimited by 12 largely convergently orientated CTCF boundary elements. To define the downstream boundary of the sub-TAD, we removed four CTCF sites in informative combinations. This showed that rather than CTCF insulators, it is the transcriptionally active alpha-globin gene that defines the downstream boundary of the sub-TAD. Further, insertion of actively transcribed fragments of the α-globin gene between the enhancers and native genes leads to a reduction in native α-globin expression and accumulation of cohesin at the insertion site. This highlights an overlap in the functional role of the fundamental elements of the genome.**

**Keywords** CTCF; Domains; Insulators; Regulation; Transcription
**Subject Category** Chromatin, Transcription & Genomics

## Introduction

Gene expression throughout development and differentiation is controlled by an interplay between three fundamental *cis*-acting regulatory elements: enhancers, promoters, and insulators. Although each type of element is classified by a working definition that enables researchers to establish the syntax of the genome, it is becoming increasingly clear that there is some overlap in the functional roles of these elements as currently defined. For example, some enhancers act as promoters (Kowalczyk et al, 2012; Mikhaylichenko et al, 2018; Nguyen et al, 2016; van Arensbergen et al, 2017; Bozhilov et al, 2021) and some promoters may also act as enhancers (Arnold et al, 2013; Dao et al, 2017; Mikhaylichenko et al, 2018; Zabidi et al, 2015). Whether enhancers and promoters can also act as insulators has been less well studied.

In mammals, insulators are frequently located at the borders of large (~50–2000 kb) regions of chromatin referred to as Topologically Associating Domains (TADs or sub-TADs; self-interacting domains that are nested within larger TADs with a median size of 185 kb) (Dixon et al, 2012; Nora et al, 2012; Phillips-Cremins et al, 2013; Rao et al, 2014). TADs are defined as regions of self-interacting chromatin, in that chromatin within a TAD has a higher contact frequency with itself than with regions in surrounding TADs (Dixon et al, 2012; Nora et al, 2012; Sexton et al, 2012). This is thought to ensure that enhancers predominantly interact with promoters present in the same TAD, adding to the specificity of gene regulation. Current models propose that TADs are formed by the extrusion of chromatin loops via translocation of the cohesin complex (Fudenberg et al, 2017; Fudenberg et al, 2016; Sanborn et al, 2015). Importantly, insulator elements recruit the zinc finger CCCTC-binding factor (CTCF), which interrupts the translocation of cohesin in an orientation-dependent manner and stabilises this protein complex on chromatin. Consistent with this model, cohesin has been shown to be enriched at active boundary elements (Hadjur et al, 2009; Parelho et al, 2008; Rubio et al, 2008; Stedman et al, 2008; Wendt et al, 2008). Deletion or inversion of insulators can alter the extent of self-interacting TADs and may enable the formation of new enhancer-promoter contacts often producing aberrant gene regulation (de Wit et al, 2015; Dowen et al, 2014; Flavahan et al, 2016; Gomez-Marin et al, 2015; Guo et al, 2015; Hanssen et al, 2017; Hnisz et al, 2016; Lupiáñez et al, 2015; Narendra et al, 2016; Narendra et al, 2015; Nora et al, 2012; Sanborn et al, 2015; Stolper et al, 2023).

[1]MRC Weatherall Institute of Molecular Medicine, Radcliffe Department of Medicine, University of Oxford, Oxford, UK. [2]Biomedical Services, Medical Sciences Division, University of Oxford, Oxford, UK. [3]The Francis Crick Institute, London, UK. [4]MRC WIMM Centre for Computational Biology, MRC Weatherall Institute of Molecular Medicine, Radcliffe Department of Medicine, University of Oxford, Oxford, UK. [5]These authors contributed equally: Lucy J Cornell, Caroline L Harrold. ✉E-mail: jim.hughes@imm.ox.ac.uk; doug.higgs@imm.ox.ac.uk; mira.kassouf@imm.ox.ac.uk

Despite this coherent model integrating the role of enhancers, promoters, and insulators relating to genome structure to gene expression, there are exceptions. For example, not all CTCF-bound sites act as insulators (Dixon et al, 2012) and, importantly, not all TADs are flanked by convergent CTCF sites (Gomez-Marin et al, 2015; Rao et al, 2014). For example, deletion of a CTCF-rich region in the *Firre* locus showed that its TAD boundary is preserved, providing evidence for CTCF-independent boundaries (Barutcu et al, 2018). Global depletion of CTCF results in a loss (Hyle et al, 2019; Nora et al, 2017) or weakening (Wutz et al, 2017) of ~80% of TADs across the genome, but not all TADs depend on CTCF. Of interest, removal of CTCF does not lead to widespread mis-regulation of gene expression (Hyle et al, 2019; Nora et al, 2017; Karpinska et al, 2025). Together, these observations suggest that elements other than CTCF insulators may also act as functional boundaries. Previous reports have proposed that actively transcribed genes may play such a role. First, Transcriptional Start Sites (TSSs) of housekeeping genes are enriched at TAD borders (Dixon et al, 2012; Hong and Kim, 2017; Bonev et al, 2017) and, second, the act of transcription can affect 3D genome structure independently of CTCF (Barutcu et al, 2019; Busslinger et al, 2017; Heinz et al, 2018; Bozhilov et al, 2021; Banigan et al, 2023; Zhang et al, 2023; Zhang et al, 2021; Barshad et al, 2023; Zhang et al, 2020; Zhang et al, 2019). Similar to the conflicting data surrounding the role of loop extrusion and CTCF in regulating gene expression, the instructive nature of PolII and transcription on interaction domain formation and gene expression programmes remains debatable (Mitchell and Fraser, 2008; Jiang et al, 2020; Hsieh et al, 2020; Zhang et al, 2021). Specifically, whether an actively transcribed gene can behave as a boundary for an interaction domain or an enhancer-blocking insulator, in a similar manner to a CTCF element, has not been thoroughly tested in mammalian systems.

The duplicated mouse α-like globin genes (*Hba-a1* and *Hba-a2*) and their five enhancers (R1, R2, R3, Rm, and R4) form a very well-characterised, small ~70 kb tissue-specific sub-TAD in erythroid cells, arranged 5'-R1-R2-R3-Rm-R4-*Hba-a1*-*Hba-a2*-3' which is nestled within a larger tissue-invariant TAD (Fig. EV1). In the past, this locus has been extensively used to establish the principles underpinning mammalian gene regulation and relating genome structure to function (Brown et al, 2018; Davies et al, 2016; Hanssen et al, 2017; Hay et al, 2016; Oudelaar et al, 2020; Oudelaar et al, 2018; Oudelaar et al, 2019). The α-globin sub-TAD is flanked by several, largely convergent CTCF binding sites (Hanssen et al, 2017). We have previously shown that in vivo deletion of two CTCF binding sites at the upstream border of the α-globin locus results in an expansion of the sub-TAD and the incorporation of three upstream genes into a newly formed sub-TAD. These three genes become upregulated in erythroid cells via interactions with the α-globin enhancers (Hanssen et al, 2017). Therefore, the intact 5' boundary normally delimits enhancer interactions and thereby contributes to tissue-specific regulation of gene expression. However, it is not known which, if any, of the regulatory elements produce a similar boundary at the 3' limit of the α-globin sub-TAD.

To investigate the insulators within and downstream of the sub-TAD, we used CRISPR-Cas9-mediated targeting to generate mouse models with mutations of four relevant CTCF binding sites. Specific CTCF sites were targeted individually and in informative combinations. We found that the 3' border of the TAD is only minimally affected by the disruption of any of the tested CTCF

sites, either individually or in combination. Rather, this border is predominantly defined by the actively transcribed downstream α2-globin gene (*Hba-a2*). In addition, we found that, when transcribed, the upstream α1-globin gene (*Hba-a1*) acts as a partial barrier to enhancer-promoter interactions with the downstream α2-globin gene. In support of this, insertion of transcriptionally active fragments of the α-globin gene between the enhancers and the endogenous genes leads to suppression of expression of the endogenous genes; this was also associated with accumulation of cohesin around the insertion sites. Together, our findings illustrate that actively transcribed genes themselves may act as insulators.

## Results

### Deletion of downstream CTCF sites (HS + 44/HS + 48) results in only minor expansion of the self-interacting α-globin sub-TAD with no changes in gene expression

We have previously characterised the sequence and orientation of 16 CTCF binding sites within and flanking the mouse α-globin locus (Hanssen et al, 2017). In general, the sub-TAD is immediately flanked by 12 convergently orientated sites (Figs. 1 and EV1). Using NG Capture-C (hereafter referred to as Capture-C) in erythroid cells, we have previously shown that two CTCF binding sites (HS + 44/ + 48) at the 3' end of the α-globin locus interact relatively weakly with the two CTCF-bound sites (HS-38/-39) that constitute the upstream (5') boundary of the sub-TAD (Hanssen et al, 2017; Hua et al, 2021). These CTCF sites do not interact at all with the active enhancer elements (R1, R2, R3, Rm, and R4) within the α-globin sub-TAD. Therefore, we initially considered that HS + 44/ + 48 might delimit the interactions of the α-globin enhancers with promoters lying downstream of the sub-TAD in a similar way to that of HS-38/-39 at the upstream border. We therefore used CRISPR-Cas9-mediated mutagenesis to generate mice with deletions in the binding sequences of these two CTCF sites (Δ44/48; Figs. 1A and EV1).

In erythroid cells, isolated from the spleens of homozygous Δ44/48 mice, mutations of the HS + 44/ + 48 binding sequences resulted in a complete loss of CTCF binding at these sites without affecting CTCF binding to other, nearby sites (Fig. 1A). In addition, other than the loss of peaks at the deleted CTCF sites, the chromatin accessibility around the α-globin locus in Δ44/48 erythroid cells remained unaltered when compared to wild-type (WT) erythroid cells, indicating that other tissue-specific regulatory elements remained intact and unaltered.

To investigate whether the removal of the HS + 44/ + 48 sites resulted in changes in local genome topology, Capture-C was performed from viewpoints across the α-globin locus in WT and Δ44/48 primary erythroid cells. Capture-C profiles from the viewpoint of the functional upstream boundary (CTCF site HS-38) show that in the absence of CTCF binding to the HS + 44/ + 48 sites, the diffuse interactions over the downstream sites had shifted to the next pair of downstream CTCF binding sites (HS + 65/ + 66), resulting in a minor expansion of the sub-TAD (Fig. EV2A). Moreover, Capture-C profiles from the viewpoint of the R2 enhancer showed that the interactions between the enhancer elements and the α-globin genes (ie the sub-TAD) are largely unchanged upon deletion of the HS + 44/ + 48 CTCF sites

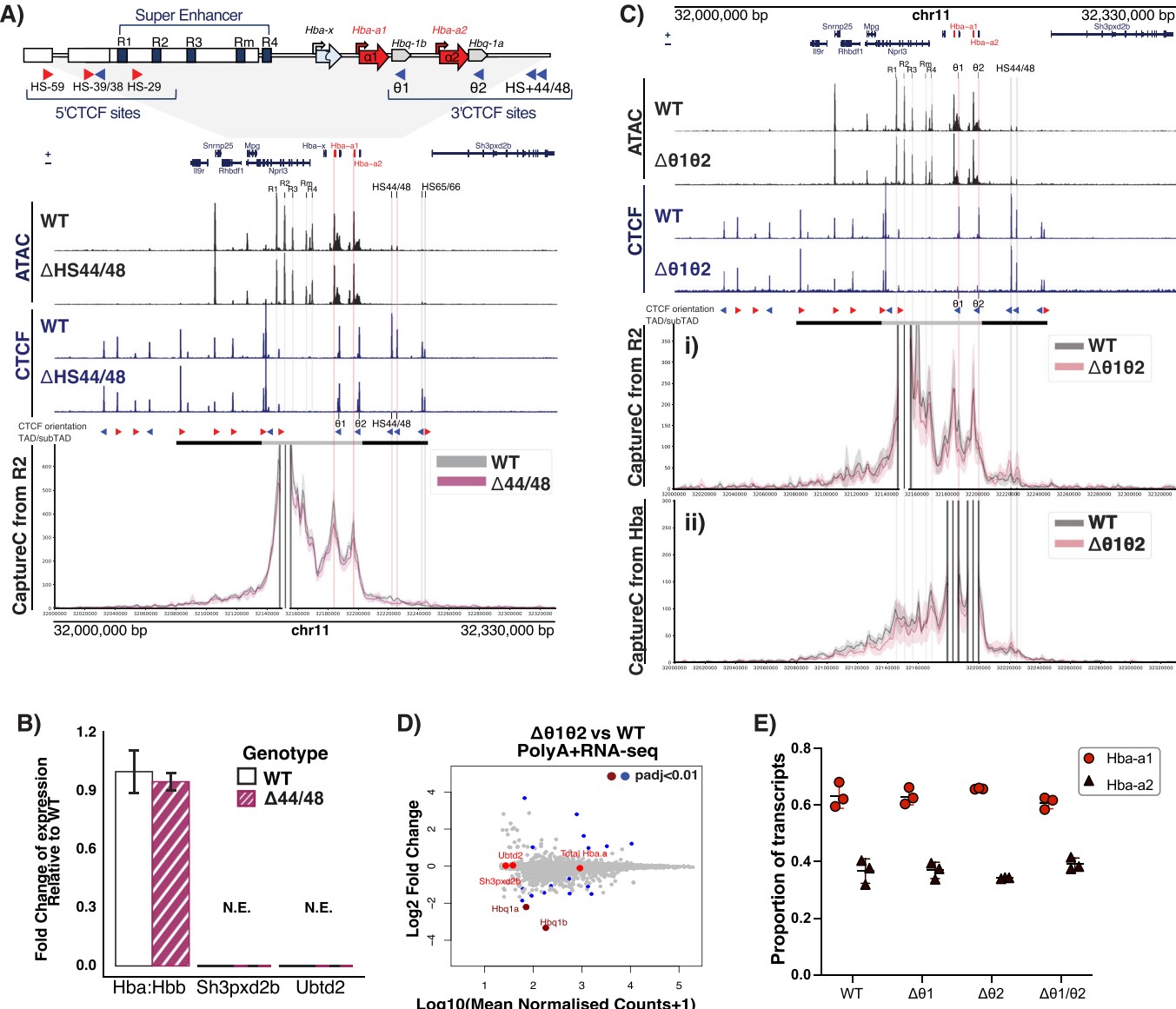

**Figure 1. Downstream CTCF-bound sites do not regulate the differential interactions and expression of the α-globin genes.**

(A) Characterisation of ΔHS + 44/HS + 48 primary erythroid cells. Top tracks show profiles for ATAC-seq (black) and CTCF ChIP-seq (navy) in primary erythroid cells (Ter119 + ) isolated from WT (Hanssen et al, 2017) and ΔHS + 44/HS + 48 mice. Profiles show normalised (RPKM) and averaged data from n = 3 biological replicates across the α-globin locus (mm9, chr11:32,000,000–32,330,000) with genes and genomic position, with positioning of genes above or below representing sense and antisense transcription, respectively. The adult α-globin genes are highlighted in red. The individual α-globin superenhancer elements (R1, R2, R3, Rm, and R4) are highlighted in grey. The horizontal grey bars between the tracks represent the ~70 kb α-globin sub-TAD (light grey, chr11:32,136,000–32,202,000) nested within a larger ~165 kb TAD (dark grey, chr11:32,080,000– 32,245,000) (Oudelaar et al, 2020). The orientation of CTCF motifs is shown under peaks by red (forward) and blue (reverse) arrows. NG Capture-C interaction profiles of the α-globin locus from the viewpoint of the R2 enhancer element with an exclusion zone around the viewpoint, in WT (grey) and ΔHS + 44/HS + 48 (purple) Ter119+ primary erythroid cells. The profiles represent normalised and averaged unique interactions from n = 3 biological replicates with halos representing ± standard deviation, smoothed with a 1D Gaussian filter. (B) Reverse transcription qPCR expression analysis. Fold changes in α- (Hba) and β-globin (Hbb) mRNA ratio, Sh3pxd2b mRNA, and Ubtd2 mRNA in ΔHS + 44/HS + 48 (purple) compared to WT (white) Ter119+ erythroid cells, normalised to 18S RNA. Mean of n = 3 biological replicates shown. Error bars display ± standard deviation. (C) Characterisation of Δθ1/θ2 primary erythroid cells. As in (A) but showing profiles from primary erythroid cells (Ter119 + ) isolated from WT and Δθ1/θ2 mice. NG Capture-C interaction profiles of the α-globin locus from the viewpoint of (i) the R2 enhancer element (ii) Hba-a1/2 genes. (D) Differential gene expression WT vs Δθ1/θ2 PolyA + RNA-seq from primary erythroid cells. MAplot of Log2 fold change in gene expression relative to WT against the Log10 read counts; each dot represents a gene. Highlighted in red are Sh3pxd2b, Ubtd2, Hba-a (representing total Hba-a1/2) and the Δθ1/θ2 associated pseudogenes Hbq1a/b. Significant changes plotted with dark red or blue (Wald test P value, Benjamini–Hochberg corrected: Padj <0.01), non-significant changes are in grey and bright red. The θ1/θ2 associated genes are downregulated upon CTCF site deletion, and total Hba-a1/2 is not significantly changed. (E) Relative expression of Hba-a1/2 mRNA. From WT, Δθ1, Δθ2 and Δθ1θ2 primary erythroid cells (Ter119 + ). Variant calling analysis performed on Poly(A) + RNA-seq data from n = 3 biological replicates revealed the percentage of reads originating from transcripts of Hba-a1 (light red) or Hba-a2 (dark red). Error bars display ± standard deviation, and each point represents a biological replicate. No significant effect of genotype on the proportion of each α-globin variant was detected by a one-way ANOVA (P > 0.05) with a Tukey post hoc test. Source data are available online for this figure.

(Fig. 1A). Consistent with this was the observation that there were no changes in local gene expression in Δ44/48 erythroid cells. The two genes directly downstream of the α-globin locus (*Sh3pxd2b* and *Ubtd2*) are not expressed in WT primary erythroid cells, and RT-qPCR analysis found no detectable difference in expression of *Sh3pxd2b*, *Ubtd2*, or *Hba-a1/2* in ΔHS + 44/ + 48 erythroid cells when compared to WT erythroid cells (Fig. 1B).

Taken together, these findings show that rather than behaving as a strong insulator, the HS + 44/ + 48 CTCF sites behave as a minor boundary to loop extrusion and the potential for chromatin interactions. However, unlike the previously characterised 5' boundary and other boundaries described in the literature (Dowen et al, 2014; Flavahan et al, 2016; Gomez-Marin et al, 2015; Hanssen et al, 2017; Hnisz et al, 2016; Narendra et al, 2016; Narendra et al, 2015; Nora et al, 2012; Sanborn et al, 2015), removal of these CTCF sites does not lead to any changes in gene expression within or flanking the α-globin gene cluster.

## Investigating the role of CTCF sites lying within the α-globin sub-TAD (θ1/θ2) in regulating gene expression

Since the HS + 44/ + 48 sites do not constitute the 3' boundary of the α-globin sub-TAD, we next considered the more proximal CTCF sites that coincide with the θ1/θ2 genes (*Hbq1b* and *Hbq1a*) (Fig. 1), which are situated inside the α-globin sub-TAD and within the duplicated region of the α-globin locus. The θ1/θ2 genes are α-like genes of unknown function. In addition to displaying diffuse interactions with the downstream HS + 44/ + 48 sites, the upstream (5') boundary of the α-globin sub-TAD also interacts with the θ1/θ2 CTCF-bound sites (Hanssen et al, 2017; Hua et al, 2021) (Fig. 1). Furthermore, we have previously shown that Capture-C interaction profiles from the viewpoint of any active regulatory element inside the undisturbed sub-TAD display a pronounced reduction in interactions immediately downstream of the 3' α2-globin gene (*Hba-a2*), which, within the resolution of these studies, appears to coincide with the θ2 CTCF binding site (Brown et al, 2018; Davies et al, 2016; Hanssen et al, 2017; Hay et al, 2016). Therefore, it seemed possible that this CTCF-bound site could act as the 3' boundary of the α-globin sub-TAD. To investigate the roles of CTCF sites inside an active sub-TAD with respect to genome structure and gene regulation, we used CRISPR-Cas9-mediated mutagenesis to generate mice with deletions of the prominent θ CTCF sites individually or in combination (Δθ1, Δθ2, and Δθ1/θ2 Appendix Fig. S1; Figs. EV3A,B and 1C).

Analysis of primary erythroid cells from Δθ1, Δθ2, and Δθ1/θ2 homozygous mice showed that mutations of the CTCF binding sequences resulted in the abrogation of CTCF binding at the specifically targeted sites (Figs. 1C and EV3A,B). We noted very small CTCF peaks that remained 5' of the targeted sites in the single deletions; these sites are not convergently orientated relative to the sub-TAD and did not appear to bind CTCF in Δθ1/θ2 (Fig. 1C); therefore, CTCF binding is sufficiently depleted at these sites. Again, there were no additional changes in chromatin accessibility around the α-globin locus in erythroid cells derived from any of the three θ mouse models when compared to WT erythroid cells (Figs. 1C and EV3A,B). To determine if there was a change in α-globin expression, PolyA + -RNA-seq was performed; on Δθ1/θ2 erythroid cells there were no significant changes in total *Hba-a1/2* expression and no changes for the downstream genes *Sh3pxd2b* and

*Ubtd2*; transcription of the θ1/θ2 associated *Hbq1a/b* pseudogenes were also downregulated as would be expected from a disruption of the associated proximal CTCF sites (Figs. 1D and EV3C,D). In addition, a differential expression profile of *Hba-a1/2* is observed (Fig. 1E), which is discussed later in the text.

To investigate whether removal of CTCF sites within and immediately downstream of the active α-globin locus caused changes to the sub-TAD structure, Capture-C was performed from the viewpoint of the α-globin promoters or R2 enhancer in WT, Δθ2, and Δθ1/θ2 primary erythroid cells. In both Δθ2 and Δθ1/θ2 erythroid cells, the 3' boundary of the sub-TAD remained largely intact, and the pronounced reduction in chromatin interactions downstream of the α2-globin gene persisted, despite the loss of CTCF binding at θ2 and both θ sites, respectively (Figs. 1C and EV3B). In addition, the overall sub-TAD structure remained largely unaffected. Hence, the θ1/θ2 CTCF binding sites are not essential to form the 3' boundary of the sub-TAD.

## The α-globin sub-TAD boundary appears to be independent of all 3' CTCF sites

Having found that sub-TAD function and structure appeared to be independent of the pairs of CTCF sites, HS + 44/HS + 48 and θ1/θ2, we next considered that the sites may act redundantly to form the 3' boundary of the sub-TAD; such that deletion of the pairs alone was not sufficient to ablate their combined function. Therefore, the Δθ1/θ2 mice were retargeted to include the ΔHS + 44/HS + 48 deletion, to create mice lacking the 4 primary 3' CTCF sites of the α-globin locus (Δθ1/θ2/HS + 44/HS + 48; Fig. 2).

CTCF ChIP-seq confirmed the loss of CTCF binding to the four sites with no additional changes in chromatin accessibility as assessed by ATAC-seq in primary erythroid cells from Δθ1/θ2/HS + 44/HS + 48 homozygous mice (Fig. 2A). In addition, heterozygous and homozygous mice had Mendelian ratios and haematology scores consistent with the normal ranges of WT mice, indicating they were phenotypically unaffected by the CTCF site deletions (Fig. 2B,C). When assessing changes in gene expression by RT-qPCR in Δθ1/θ2/HS + 44/HS + 48 mice, there was once again no significant change in the expression of α-globin, or the 3' genes (Fig. 2D), similarly observed in ΔHS + 44/ + 48 and Δθ1/θ2 (Fig. 1B,D).

To determine if there were any differences in sub-TAD interactions with the loss of the Δθ1/θ2/HS + 44/HS + 48 CTCF sites, Capture-C was performed on primary erythroid cells (Figs. 2A and EV2B). Other than the expected loss of signal over the deletion sites and very small gain of interactions to the next available CTCF-bound sites (HS + 65/66), the major interactions between the enhancer elements (R2; Fig. 2A) and the α-globin genes (Fig. EV2B) were unaffected by the loss of the 4 CTCF sites (θ1/θ2/HS + 44/HS + 48).

When observing interactions with the R2 enhancer element across all of our CTCF site deletion models (ΔHS + 44/HS + 48; Fig. 1A, Δθ1/θ2; Fig. 1C, Δθ1/θ2/HS + 44/HS + 48; Fig. 2A) it becomes clear that the enhancer-promoter interactions that form the strongly interacting sub-TAD are largely independent of the CTCF sites and that the 3' boundary of the sub-TAD is in fact marked by the active α2 globin gene itself, exemplified by a sharp drop in enhancer interactions at *Hba-a2* (Fig. 2A, black arrow).

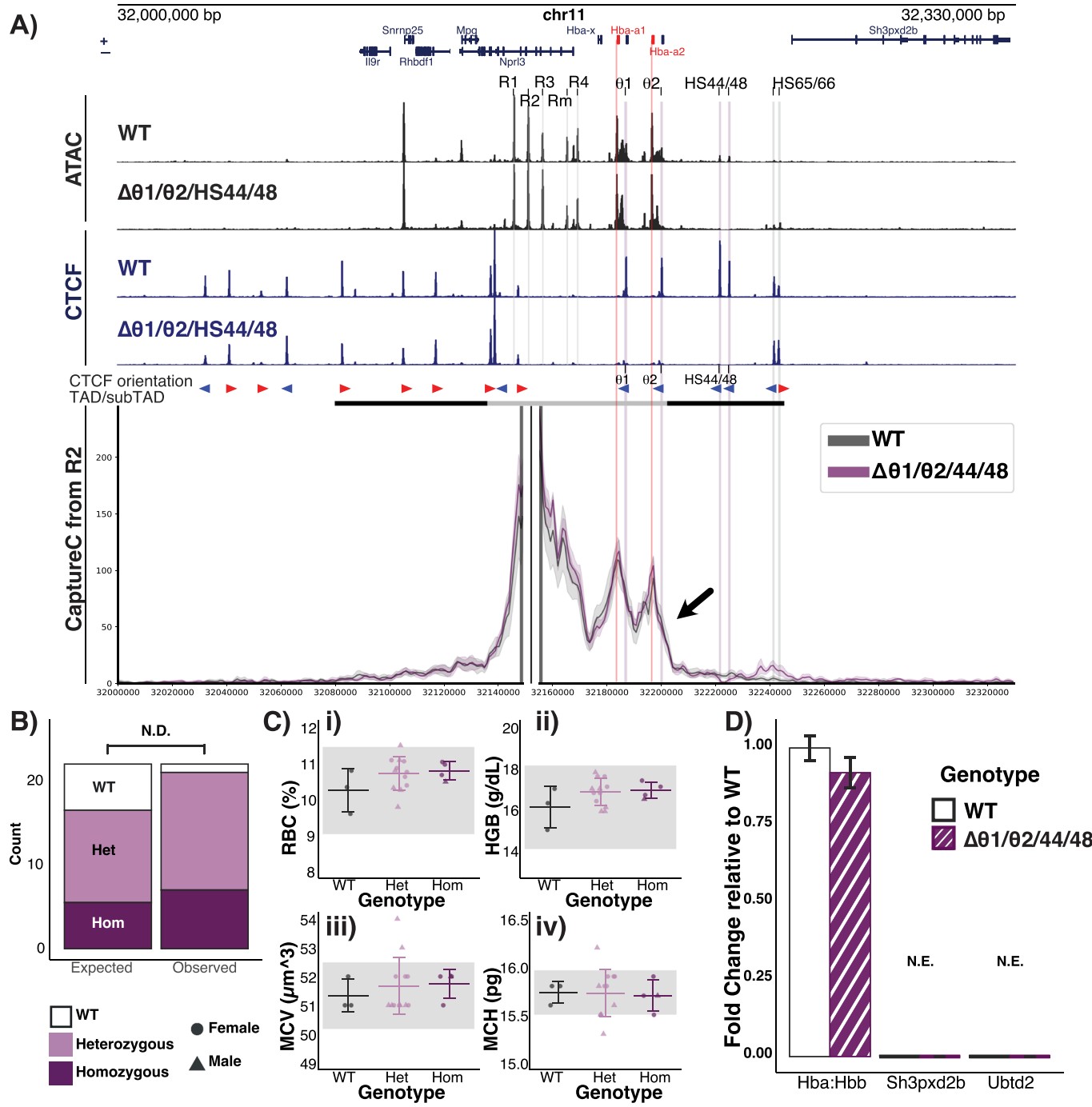

The actively transcribed α2-globin gene acts as the downstream boundary of the α-globin sub-TAD

Although the CTCF sites lying close by and in between the α-globin genes play no role in forming the 3' boundary of the α-globin sub-TAD, we considered whether they might play a role in fine-tuning gene expression within the sub-TAD.

In humans, the upstream 5' α-globin gene (*HBA2*; the equivalent of mouse *Hba-a1*) is closer to the enhancers and is more highly expressed than the downstream 3' α-globin gene (*HBA1*; the

equivalent of mouse *Hba-a2*). The *HBA2* gene produces ~70% of the total α-globin mRNA whilst the distal *HBA1* contributes only ~30% (Liebhaber et al, 1985; Liebhaber and Kan, 1981; Orkin and Goff, 1981). To note, the promoters of the two human HBA genes are only separated by ~3.7 kb and are identical in sequence, hence the disparity in transcriptional output is surprising, as these two genes would be expected to be under similar control by the enhancers. To determine if this differential expression of the α-globin genes also holds true for mouse, Poly(A) + RNA-seq was performed on primary WT erythroid cells. The presence of a single

**Figure 2.  Characterisation of deletion of multiple CTCF-bound sites downstream of the α-globin locus.**

(A) Characterisation of Δθ1/θ2/HS + 44/HS + 48 primary erythroid cells. Top tracks show profiles for ATAC-seq (black) and CTCF ChIP-seq (navy) in primary APH-treated spleen cells isolated from WT (Hanssen et al, 2017) and Δθ1/θ2/HS + 44/HS + 48 homozygous mice. Profiles show normalised (RPKM) and averaged data from $n = 3$ biological replicates (ATAC) and $n = 2$ biological replicates (CTCF) across the α-globin locus (annotations are as in Fig. 1A). NG Capture-C interaction profiles of the α-globin locus in primary erythroid (Ter119 + ) cells from the viewpoint of the R2 enhancer element with an exclusion zone around the viewpoint, in WT (grey) and Δθ1/θ2/HS + 44/HS + 48 (purple). The interaction profiles represent normalised and averaged unique interactions from $n = 3$ biological replicates and halos representing ± standard deviation, smoothed with a 1D Gaussian filter. The black arrow highlights the apparent boundary of the sub-TAD. (B) Mendelian ratios of Δθ1/θ2/HS + 44/HS + 48 mice litters. Counts do not significantly differ from an expected Mendelian ratio, as calculated by the Chi-squared Goodness of Fit test ($n = 22$, $X2 = 4.9091$, d.f. $= 2$, $P$ value $= 0.0859$). (C) Haematology and blood counts of WT and Δθ1/θ2/HS + 44/HS + 48 mice, showing (i) normal red blood cell counts (RBC), (ii) haemoglobin (HGB), (iii) mean corpuscular volume (MCV) and (iv) mean corpuscular haemoglobin (MCH) from WT (black), Δθ1/θ2/HS + 44/HS + 48 Heterozygous (light purple) and Homozygous (dark purple) mice. No significant effect of genotype was detected by two-way ANOVA ($P > 0.05$). Error bars represent standard deviation, and each point represents a biological replicate; $n = 19$; WT: $n = 3$, Het: $n = 12$, Hom: $n = 4$. Grey shading indicates the normal range per parameter, calculated as ± two standard deviations from the WT mean. (D) Reverse transcription qPCR expression analysis of α- and β-globin mRNA ratio, Sh3pxd2b mRNA, and Ubtd2 mRNA in WT (white) and homozygous Δθ1/θ2/HS + 44/HS + 48 (purple) Ter119+ erythroid cells, normalised to 18S RNA. Mean of $n = 3$ biological replicates shown, error bars display ± standard deviation. Data normalised to WT. Source data are available online for this figure.

SNP in the third exon of *Hba-a1* and *Hba-a2* (mm9: chr11: 32184343 T/C) allows for variant calling analysis of the reads originating from the α-globin transcripts. As in human, *Hba-a1* (the gene closest to the enhancers) accounted for ~66% of the total α-globin mRNA and *Hba-a2* (the distal gene) accounted for ~34% (Fig. 1E).

Consistent with its higher level of expression, we have previously shown that the *Hba-a1* promoter also preferentially interacts with the α-globin enhancers relative to the *Hba-a2* promoter (Davies et al, 2016) (Fig. EV2B, iii, iv, v). Both the preferential expression and interactions with the more proximal *Hba-a1* could be due to the presence of the intervening CTCF sites (θ1/θ2) between the two *Hba* genes (in the order 5'-α1-θ1-α2-θ2-3'). We therefore investigated whether the θ CTCF sites, situated in between the α-globin genes, regulate the differential expression of the mouse α-globin genes and/or their interactions with the α-globin enhancers.

First, Poly(A) + RNA-seq was performed on primary erythroid cells isolated from Δθ1, Δθ2, and Δθ1/θ2 mice and used the variant calling analysis described above on the α-globin transcripts. In all three models, the relative proportions of transcripts produced by *Hba-a1* and *Hba-a2* were similar to that of WT (Fig. 1E), showing that the loss of CTCF binding between and downstream of the α-globin genes did not affect the preferential expression of *Hba-a1*.

Next, to investigate whether the loss of CTCF binding around the α-globin genes altered differential interactions of *Hba-a1/2* with the enhancers, from the Capture-C data analysed from the viewpoints of the α-globin promoters in WT and Δθ1/θ2/HS + 44/HS + 48 primary erythroid, separate interaction profiles were generated for the *Hba-a1* and *Hba-a2* promoters following previously described analysis (Davies et al, 2016). When erythroid cells from Δθ1/θ2/HS + 44/HS + 48 were analysed, the profiles were consistent with WT, hence the same differential interaction profiles of the α-globin promoters were observed as previously reported in WT erythroid cells (Fig. EV2B, iii–vi), indicating that the differential interactions of *Hba-a1/2* with the enhancers is not influenced by presence or absence of the Δθ1/θ2 CTCF sites.

Therefore, in summary these findings suggest that CTCF binding at θ1 and/or θ2 do not regulate the differential interactions of *Hba-a1/2* with the α-globin enhancers or the preferential expression of the proximal α1-globin gene (*Hba-a1*) compared to the distal α2-globin gene (*Hba-a2*). Rather, it appears that the transcribed α1-globin gene may act as a partial boundary to the α2-globin gene in terms of access to a shared set of enhancers just as

the α2-globin gene acts as the downstream boundary of the sub-TAD.

## Active transcription from ectopically inserted gene fragments acts similarly to insulator elements

To test the hypothesis that the α-globin genes may themselves be forming the primary sub-TAD boundary, with *Hba-a1* also potentially acting as a partial boundary to *Hba-a2*, we used the α-globin sub-TAD structure to design a boundary activity assay whereby test sequences are inserted between the cluster of α-globin enhancers and the α-globin genes enabling us to test their potential to act as insulators (Tsang et al, 2024) (Fig. 3A). Insertion of an insulating sequence between the enhancer elements and the native α-globin genes, would intercept existing enhancer-promoter interactions and reduce expression of the α-globin genes, as measured in this assay by the *Hba-a1::mVenus* reporter fluorescence from in vitro-derived erythrocytes (CD71+ve) (Francis et al, 2022). We have recently used this to determine the relative insulation strengths of different CTCF site sequences (Tsang et al, 2024).

Whilst CTCF is considered the primary insulator element in mammalian genomes, transcription has been also been associated with domain boundaries (Bonev et al, 2017; Dixon et al, 2012); reported to interact with the cohesin loop extrusion machinery (Busslinger et al, 2017); and with disruption of expression of downstream genes (Bozhilov et al, 2021; Cho et al, 2018). We therefore considered that transcription from the α-globin genes was a possible cause of the boundary/insulation effect seen at the 3' end of the locus.

To functionally test the insulator activity of the α-globin genes, fragments were designed to insert into the activity assay: αP$_{Only}$, the isolated α-globin promoter (373 bp: spanning the 5'UTR TSS + 31 bp to TSS -342bp); αG$_{Only}$, the isolated α-globin gene body (774 bp: spanning TSS + 37 bp to the PolyA signal); and αP + G, the α-globin promoter and gene body together (1.1 kb) (Appendix Fig. S2 and Appendix Table S2). These regions in their native context do not bind CTCF (see Fig. 1A CTCF ChIP-seq, where there is no signal directly over *Hba-a/2*) and do not contain a consensus CTCF binding motif (JASPAR, threshold of $P < 1e-4$) and so any insulator effect would not be due to CTCF binding. As controls, the HS-38 CTCF site was inserted, which is known to be a strong insulator element and the 5' boundary of the α-globin locus (Hanssen et al,

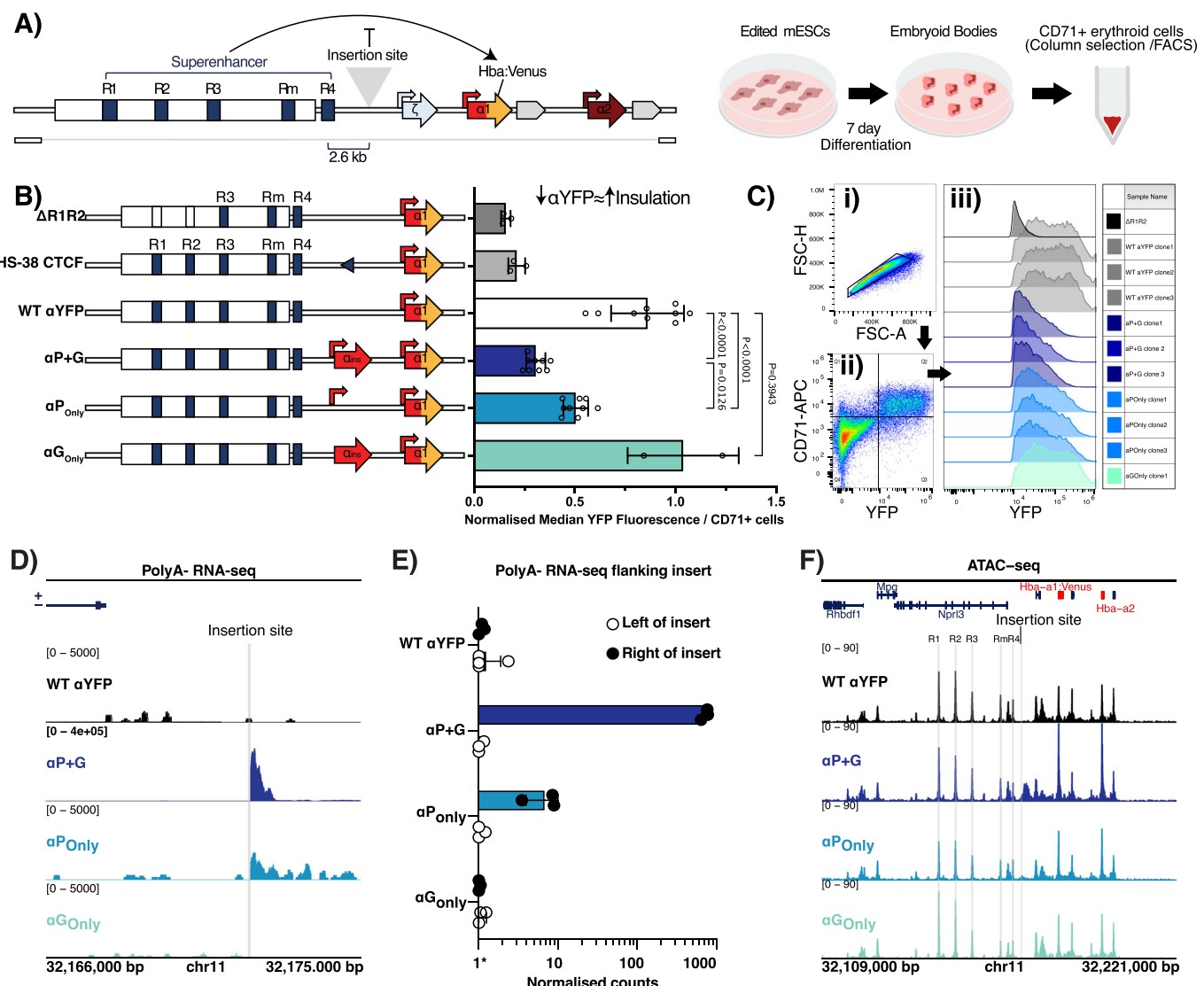

**Figure 3. Insulation strength of α-globin gene fragments in a boundary reporter assay correlates with their transcriptional outputs.**

(A) Schematic of the design of the insulator assay. The α-globin locus was edited in Mouse Embryonic Stem Cells (mESCs). The key features of the design are: (1) A single insertion site is located between the globin genes and the enhancer elements; (2) Hba-a1 is tagged with mVenus (red/yellow) to allow readout of α-globin expression by FACS; (3) the cell line is hemizygous for the α-globin locus to allow for increased targeting efficiency and single allele genomics; finally, (4) engineered mESCs are differentiated and erythroid cells are isolated (CD71+ve) via a 7-day in vitro differentiation protocol allowing for gene expression read out (Francis et al, 2022).
(B) Insulation strength of α-globin gene fragments in a boundary reporter assay. Left: Schematics of the models: ΔR1R2 = mESC with deleted R1 and R2 enhancers and with αYFP; HS-38 = positive control/mESC with HS-38 CTCF site inserted; WT αYFP = parental αYFP reporter with no insert (n = 3); αP + G = inserted α-globin promoter and gene body (n = 3); αP$_{Only}$ = inserted α-globin promoter only (n = 3); αG$_{Only}$ = inserted α-globin gene body only without the promoter (no expected transcription) (n = 1). Right: Median YFP fluorescence in the CD71+ fraction of cells measured by FACS and normalised to the parental αYFP reporter. Data acquired from clones (numbers indicated above) across three independent differentiations. Error bars display standard deviation. Statistical analysis was performed using a one-way ANOVA with a Tukey post hoc test, P values are presented above comparisons. (C) FACS gating strategy and density plots of YFP fluorescence in the CD71+ fraction of cells. (i) Single cells were gated by FSC-H/A, (ii) then gated for CD71 + /YFP + . (iii) Density plots for each clone from one differentiation shown to the left: ΔR1R2 (dark grey), WT αYFP (grey), αP + G (dark blue), αP$_{Only}$ (blue) and αG$_{Only}$ (teal). (D) RPKM normalised PolyA- RNA-seq derived from CD71+ erythroid cells from each of the indicated models, aligned to the Hba-a1-only custom genome. Bigwigs are visualised on the sense strand as mean of data from n = 3 independent differentiations of a representative clone per genotype. Note that the reference does not have inserted sequences and should be used as an indication for the activity around the inserted sites. (E) Quantification of PolyA- transcription from either the left or the right-hand side of the insert, using the PolyA- RNA-seq signal (as in D) in each of the models. Normalised counts were generated using Featurecounts and DEseq2. *Pseudo count of 1 was assigned to counts of 0 to allow for visualising on a logarithmic scale. For changes in expression in the region right of the insert, WT αYFP-αP$_{Only}$, Padj = 0.00016 and for WT αYFP-αP + G, Padj= 3.69e-48. (F) RPKM normalised ATAC-seq performed on CD71+ erythroid cells derived from each of the indicated engineered models and aligned to a custom mm9 genome with Hba-a1::mVenus. Note that the reference does not have inserted sequences in the reference and should be used as an indication for the activity around the insert sites—further the signals over Hba-a1/2 are an amalgamation of reads from the native and inserted variants. The insertion site is highlighted in grey and the Hba-a1::mVenus reference in red. Data are from each of the clones shown in (B, C) and (where possible) are averages of biological replicates. Source data are available online for this figure.

2017; Tsang et al, 2024), and also used an enhancer-deficient reporter (ΔR1R2) to determine the lower bounds of the expression assay. To test for the insulation activity, correctly targeted mESCs were in vitro differentiated into embryoid bodies (EBs). α-YFP fluorescence measured in the CD71 + /YFP+ cell population of cells with αP + G inserts revealed a significant decrease in α-YFP compared to that in the parental reporter erythroid cells (WT α-YFP), indicating the ectopic gene interferes with the expression of the downstream endogenous α-globin genes; this effect was comparable to that observed when a strong CTCF site (HS-38) was present (Fig. 3B).

The insulation observed following insertion of the α-globin gene (αP + G) may be due to several factors, including the transcription itself (transcription initiation, pausing or elongation) or intrinsic properties of the gene-body sequence. To disentangle these factors, the boundary effect of the α-globin promoter sequence only (αP_{Only}) and the α-globin gene sequence without its promoter (αG_{Only}) was tested. CD71+ erythroid cells derived from mESCs with the insertion of the gene body alone (αG_{Only}) had high α-YFP fluorescence, comparable to that of the parental reporter cells, indicating the gene sequence alone does not have any insulator-like properties, whilst cells with the promoter sequence alone (αP_{Only}) expressed lower α-YFP compared to WT α-YFP, indicative of insulator activity (Fig. 3B,C). To summarise, the insulation effect was proportionate to the transcriptional potential of the inserted sequences; αP + G inserts had the greatest decrease of αYFP reporter expression, followed by the αP_{Only} insert with the gene body alone (αG_{Only}) having a negligible effect.

To verify the transcriptional output from the inserted fragments, PolyA-specific RNA-seq was performed on in vitro-derived CD71+ cells from each of the models, allowing quantification of immature (PolyA-) transcripts from around the insertion site and mature transcripts (PolyA + ). αP + G generated many transcripts around the insertion site, with some immature transcripts firing from the αP_{Only} insert; these transcripts were exclusively directed downstream (right) of the insertion site in the expected direction of transcription (Fig. 3D,E). As expected, no transcripts were observed for the αG_{Only} inserts. In PolyA+ RNA-seq, *mVenus* transcripts reflected the trend of YFP fluorescence, and *Hba-x* was also downregulated (LFC = -0.73, BH corrected $P$adj = 5.5e-04) in the αP + G insertion model, consistent with this insertion having the greatest insulation affect (Fig. EV4A–E). Remarkably, despite already high levels of α-globin expression, total *Hba-a* counts increased significantly when the αP + G insert was added (LFC = 2.1, BH corrected $P$adj=1.38e-08) (Fig. EV4A). These additional α-globin transcripts originate from the inserted gene as highlighted by SNP-specific counts distinguishing the inserted αP + G from the native *Hba-a1/2* (Fig. EV4F); this increase is consistent with an expected increase in activity due to its increased proximity to the enhancers.

This was reflected in the accessibility of the region, with a large region of accessibility generated in the αP + G model, lesser in αP_{Only} and no gain in αG_{Only}; the rest of the cis-regulatory elements in the locus remained unaffected (Fig. 3F). Together this data highlights a correlation between transcription level and the insulator strength of the inserted gene fragments, with greater transcription leading to greater apparent insulation strength.

## Active transcription is associated with the accumulation of cohesin

The reduction in endogenous *Hba-a1/2* gene expression in this reporter assay is considered as a measure of insulator strength. However, the insulation effect observed following insertion of the α-globin gene can be interpreted by two mechanistic models. The first, is promoter competition whereby the ectopic promoter (αP + G and αP_{Only}) may compete with the native α-globin promoters for the enhancers. Alternatively, the reduction of native gene expression could be caused by a blockade to the linear progression of loop extrusion machinery by the transcriptional machinery of the insert and hence form a partial physical boundary which hinders contact between the enhancers and their cognate promoters.

To test the latter, cohesin (Rad21) ChIP-seq was performed on CD71+ erythroid cells derived from edited reporter cells with the greatest insulation effect (αP + G and αP_{Only}) (Fig. 4A). Cohesin distribution was comparable across the α-globin locus, with cohesin peaks detected at the enhancer elements, CTCF sites and active promoters; however, when looking at the changes surrounding the insertion site, there was an increase in cohesin accumulation in the regions flanking the inserts when compared to the parental reporter cell line (Fig. 4A,B). The greatest increase in cohesin is seen at the αP + G insert, with a small additional accumulation seen downstream of the αP_{Only} insert. Consistent with this, when looking at the accumulation of cohesin genome-wide and stratifying against non-CTCF-associated TSSs of active and inactive genes (RNA-seq RPKM > 1, H3K4me3 positive, no CTCF within +/−1kb), cohesin is almost exclusively found at active TSS (Fig. EV4G), consistent with other reports (Hua et al, 2021; Busslinger et al, 2017).

Since the inserted fragments appeared to be acting as insulator elements, preventing correct interaction with the endogenous target genes, we expected a gain of interactions between the enhancers and insertion site and a possible reduction of interactions with the native promoters. To determine if there was such a change in interaction profiles, Capture-C was performed from the viewpoints of the R1 and R2 enhancer elements in the CD71+ reporter cells (Fig. 4C; Appendix Fig S3). At the resolution of this assay, the enhancer interaction profile across the locus in each model was largely unchanged. In addition, due to the reads from the inserted and native α-globin reporters mapping equally to the same DpnII fragment, it was not possible to assign reporters to their source; hence, read coverage over the native *Hba-a1/2* sites originates from both native and inserted copies. Comparison of normalised interactions at fragment resolution between models did show a trend toward an increase in interactions between the R2 enhancer and the insert site in the inserted αP + G model, and a reciprocal decrease in interactions with *Hba-x* gene and near the upstream of the *Hba-a1* promoter. (Limitations of this are discussed in Appendix Supplementary information).

With this limitation in mind, the evidence that there is a gain of accessibility at the site of insertion, increased transcription and a reduction in native gene expression all indicates inserted sequences are impacting normal enhancer-promoter communication, and this may be in part due to transcription-dependent stalling of cohesin.

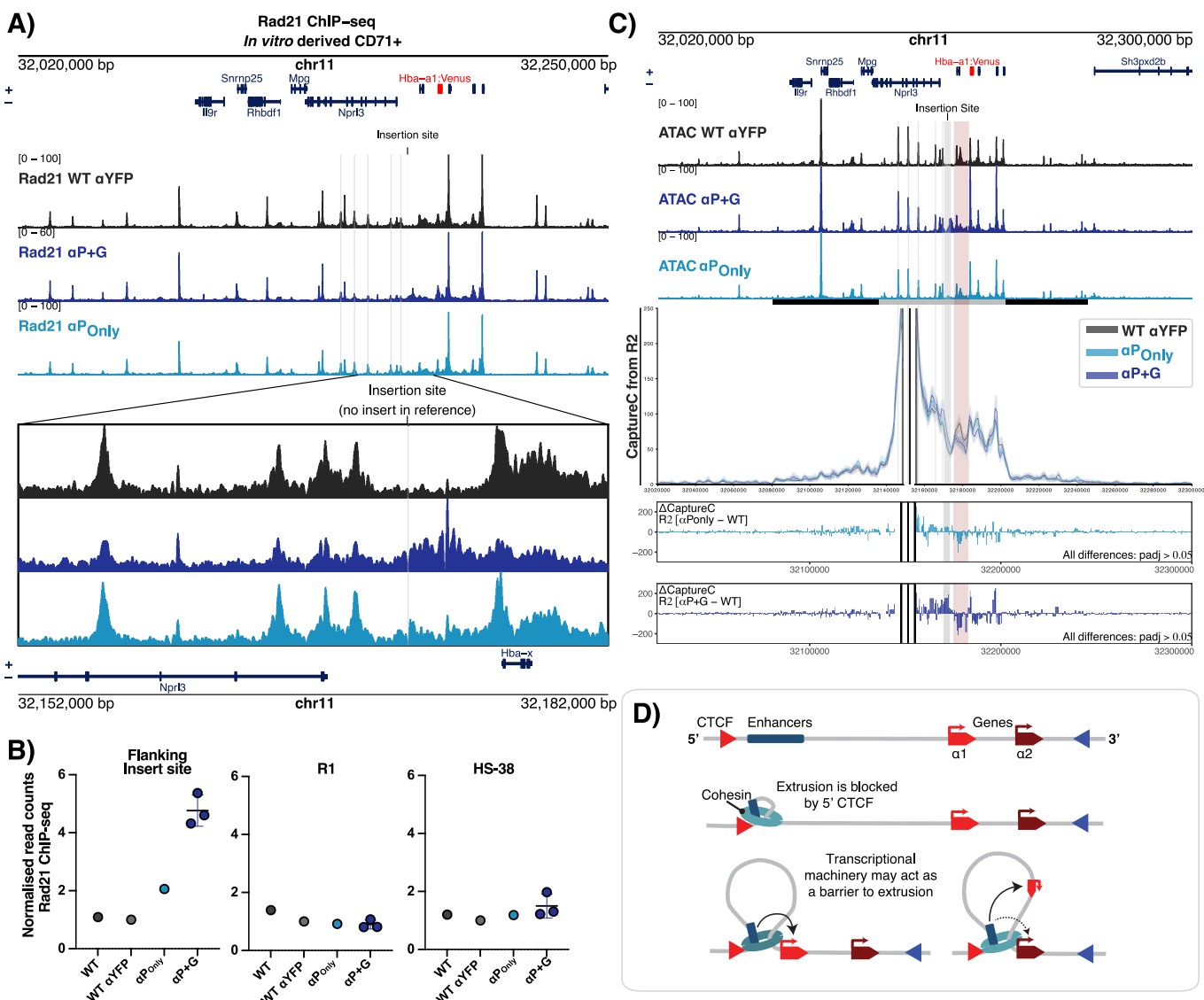

**Figure 4. Insulation strength and transcriptional outputs of α-globin gene fragments correlate with the level of cohesin accumulation.**

(A) Cohesin accumulation across the α-globin locus in CD71+ erythroid cells derived from edited reporter cells. Rad21 ChIP-seq performed on CD71+ erythroid cells derived from each of the indicated engineered models and aligned to a custom genome with *Hba-a1::mVenus* in its native position, based on the mm9 genome reference. Note that the reference does not have insert sequences in the reference and should be used as an indication of the activity around the insert sites. The insertion site is highlighted in grey and the *Hba-a1* reference in red. Top panel shows the locus (custom mm9: 32,020,000–32,250,000) whilst the bottom panel zooms in on the insertion site at the locus (custom mm9: 32,152,000–32,182,000). Bigwigs are visualised from $n = 1$ (WT-hemizygous, WT αYFP, αP$_{Only}$) and as an average of $n = 3$ (αP + G) independent differentiations of a representative clone per genotype. Where appropriate, error bars display ± standard deviation. (B) Rad21 ChIP-seq read counts in each replicate. Read counts performed over regions covering the insert site (chr11:32,171,181–32,174,582), R1 enhancer element (chr11:32,144,849–32,146,726) and HS-38 CTCF site (chr11:32,136,773–32,137,613) from the data in (A). Reads were normalised to the number of reads over a selected region in the β-globin locus and normalised to the WT αYFP to exemplify differences between models. (C) Capture-C from R2 enhancer in CD71+ erythroid cells derived from edited reporter cells. ATAC-seq in the top panel to show the positions of the elements of interest. CaptureC profiles represent the mean number of normalised unique interactions per restriction fragment from $n = 3$ (WT αYFP), $n = 4$ (αP$_{Only}$ and αP + G) independent differentiations, these are represented in dark grey, light blue and dark blue, respectively. Halos represent ± standard deviation. Differential tracks (ΔCaptureC) show subtractions (αPOnly - WT αYFP and αP + G- WT αYFP) of the mean number of unique interactions per restriction fragment, scaled to a total of 100,000 interactions in cis. Note differences represented are not determined to be significant (threshold *P*adj>0.05). The region around the insert (mm9: Insert region: chr11:32,169,922–32,173,392) is highlighted in grey, and region covering native targets (*Hba-x* - upstream of *Hba-a1* chr11:32,175,105–32,182,970) is highlighted in red. To note, read coverage over the *Hba-a1/2* genes is a combination of both inserted and native sources. (D) Schematic to show a dynamic, directional tracking mechanism of chromatin loop extrusion by cohesin from the α-globin enhancers to the promoters. Multi-protein complexes recruited to the actively transcribing genes stall cohesin translocation on chromatin, resulting in cohesin retention at active genes, in addition to CTCF binding sites. Source data are available online for this figure.

# Discussion

In many ways, the relatively small ~70 kb sub-TAD containing the mouse α-globin locus is typical of other tissue-specific TADs or sub-TADs seen in mammalian genomes. The cluster of erythroid-specific enhancer-like elements, which fulfil the definition of a superenhancer (Hay et al, 2016; Blayney et al, 2023), and the promoters of the α-like genes, are flanked by largely convergent CTCF binding sites, which are often considered to act as the structural and functional boundaries of TADs. Current models relating genome structure to function propose that TADs are formed by the extrusion of chromatin loops as the cohesin complex translocates throughout the TAD (Fudenberg et al, 2017; Fudenberg et al, 2016; Sanborn et al, 2015). The continuous process of extrusion at some point brings together all sequences within the self-interacting TAD, including the enhancers and promoters, potentially providing the proximity thought to be required for the activation of transcription. The borders of the TAD are created when cohesin is stalled and stabilised by its interaction with the N-terminal region of CTCF. Our previous studies of the mouse α-globin sub-TAD using chromosome conformation capture (Brown et al, 2018; Davies et al, 2016; Hanssen et al, 2017; Hay et al, 2016; Oudelaar et al, 2020; Oudelaar et al, 2018; Oudelaar et al, 2019) and super-resolution imaging (Brown et al, 2018) are entirely consistent with this model involving the interplay between enhancers, promoters, and insulators. Here, we have identified the elements responsible for forming the boundaries of the sub-TAD and contributing to the differential interactions between the enhancers and the promoters. Surprisingly, we find that the transcriptionally active α-globin genes rather than the anticipated CTCF binding sites play a role as boundary elements within the locus and in creating the downstream (3′) boundary of the sub-TAD.

We have previously characterised the upstream 5′ boundary of the α-globin sub-TAD (Hanssen et al, 2017), which is marked by two convergent CTCF binding sites (HS-38/-39 in Fig. 1). Deletion of these sites leads to extension of the sub-TAD to a more distal flanking CTCF site (HS-59) and erythroid-specific activation of three genes incorporated into an extended sub-TAD. Presumably, this occurs because the cohesin complex can now translocate beyond these sites. Deletion of the four CTCF sites (θ1/θ2/HS + 44/ + 48) flanking the 3′ end of the α-globin locus, which interact with the upstream boundary, behaved differently from those at the 5′ boundary. Upon deletion, there is no change in α-globin expression nor any increase in expression of downstream genes (Sh3pxd2b and Ubtd2). Deletion of these sites led to only a very small increase in the levels of interaction with the downstream flanking region beyond these sites, extending to the next CTCF sites (HS + 65/ + 66) and had no effect on the differential interactions of the two α-globin genes. In effect, removal of the θ1/θ2/HS + 44/ + 48 sites caused no change in the major transition (black arrow in Fig. 2) between interacting and non-interacting chromatin. This major boundary coincides with the actively transcribed α2-globin gene, and as the CTCF binding at θ1/θ2 is largely ablated in these deletions, this boundary is likely independent of CTCF. Upon θ1/θ2 deletions, there was also no effect on the preferential expression of the proximal α1-globin gene. This together suggests that the α1-globin gene itself acts as a partial boundary between the α2-globin gene and the α-globin genes together correspond to the prominent 3′ boundary of the sub-TAD. Consistent with our findings here our

previous observation that when human alpha-globin genes are lost, NME4 positioned ~100 kb downstream is upregulated, further supporting that the genes may also form the downstream boundary in human erythrocytes (Lower et al, 2009). In addition, similar observations were recently made at the Sox2 locus. When a reporter gene is randomly relocated across the locus, the endogenous Sox2 gene restricts its enhancers' activity and appears to act as the boundary of a functional domain created with the enhancers (Eder et al, 2025). In a different report, a reduction in native Sox 2 gene expression reverse-correlated with the strength of promoter sequences and length of their corresponding transcripts produced when inserted between the Sox 2 superenhancer and the endogenous gene (Koska et al, 2025).

Together, our findings on flanking and internal CTCF sites (Figs. 1 and 2) and the functional testing of transcribing fragments in our insulator reporter assay (Figs. 3 and 4) support the proposal that some highly active transcribed genes, in a particular context, themselves behave as functional insulators. One mechanism by which this could occur is via competition between promoters and a shared enhancer in a situation where there is unconstrained chromatin looping. In a competition model, it would propose that the promoter of the α1-globin gene outcompetes that of the α2-globin gene for access to the α-globin enhancers in which all enhancer elements appear to act as a single entity (Oudelaar et al, 2018). We cannot rule this model out from our data; indeed, the inserted α-globin gene is expressed to a far greater level than the native genes, likely due to the increased proximity to the enhancer cluster, and it is feasible that a massively transcribing gene would sequester transcriptional machinery from other neighbouring targets. This machinery could include Mediator (Ramasamy et al, 2023) and other cofactors associated with bridging between elements, such as Ldb1 (a component of the erythroid pentameric complex) (Aboreden et al, 2025) and YY1 (Weintraub et al, 2017). Similar examples of promoter competition have also been proposed in which active promoters are located between an enhancer and another, more distal promoter causing reduced activity of the distal promoter (Bartman et al, 2016; Cho et al, 2018; De Gobbi et al, 2006; Wijgerde et al, 1995; Bozhilov et al, 2021; Ealo et al, 2024; Koska et al, 2025). In addition, insertion of a PGK-Neo cassette into Hba-x led to a decrease in Hba-a1/2 expression in mice (Leder et al, 1997), which is in line with the data presented here, that an ectopically highly transcribed gene can interfere with normal Hba-a1/2 expression. However, when considering the preferential expression of Hba-a1 over Hba-a2, in the context of a freely interacting chromosomal loop, the promoters are located at a relatively similar distance from the enhancer region and are identical in sequence; no difference in expression between the two would be expected. Equally, this competition model would predict that the αP$_{Only}$ insertion into the reporter assay should have equivalent insulation strength to that of the αP + G insert, however this is not observed; αP + G is a stronger insulator than αP$_{Only}$ (Fig. 3B). The αG$_{Only}$ insert has no insulation effect, and hence the gene sequence itself carries no inherent insulator activity so the boosted insulation of the αP + G relative to the αP$_{Only}$ insert can be linked to the increased transcriptional machinery loaded at αP + G.

We therefore consider a directional tracking mechanism from enhancers to promoters, both of which have enriched levels of cohesin. Such a model of directional loop extrusion has been proposed to explain interacting stripes seen in Hi-C maps (Vian et al, 2018). We have also observed evidence that cohesin is important in the context of this locus and this insertion site, as we

observe a clear dependence of insulator strength on the orientation of CTCF sites (Stolper et al, 2023; Tsang et al, 2024). In this case, the anchor of the extruding loop would correspond to the HS-38/-39 sites, with a large proportion of cohesin travelling unidirectionally, bringing the enhancers to each gene copy progressively forming the sub-TAD, within the context of the larger CTCF-CTCF-delimited TAD. Due to the presence of transcription machineries at each actively transcribed gene, cohesin translocation is stalled and retained at each sequential gene copy, reducing the probability of enhancer interaction at more distal genes (Fig. 4B). This would be consistent with observations of the orthologous human and sheep α-globin clusters in which, the proximal duplicated α-globin gene is also expressed in preference (~70%) to the more distal gene (~30%) (Liebhaber et al, 1985; Liebhaber and Kan, 1981; Orkin and Goff, 1981; Vestri et al, 1994). Importantly, the α-globin genes and promoters are almost identical in sequence, ruling out inherent sequence-specific effects between copies, and in humans, there is no CTCF binding site between the α-globin genes, hence no intervening insulation. Indeed, there are numerous other examples of multiplications in globin loci with each additional, sequential gene providing a smaller contribution to globin mRNA and protein in a gradient; in humans (Cook et al, 2006; Gu et al, 1987; Higgs et al, 1980), in sheep (Vestri et al, 1994) and in duplications of γ-globin genes (Shimasaki and Iuchi, 1986; Huisman et al, 1977). Furthermore, in humans, a deletion of the proximal α2-globin gene (-α/) increases the output of the distal α1-globin gene from 30 to 50% (Liebhaber et al, 1985). By contrast, when there is an inactivating coding mutation in the proximal α2-globin gene (α$^M$α/), leaving its promoter and transcription intact, RNA expression from the distal α1-globin gene remains at 30% leading to the severe phenotype seen in patients with such non-deletional mutations (Harteveld and Higgs, 2010).

In the tracking model, highly transcribed genes may form a partial barrier to loop extrusion, due to the accumulation of large amounts of transcriptional machinery and regulatory factors (Brandão et al, 2019). In these scenarios, cohesin may be prevented from extruding chromatin loops due to the size of multi-protein complexes, which may be acting like 'roadblocks' via a passive blocking mechanism (Fig. 4D). Evidence that supports this comes from structural studies looking at the interaction between CTCF and cohesin. The N-terminus of CTCF structurally interacts with cohesin, however, it appears that its role is to stabilise cohesin on chromatin; when the N terminus is mutated, cohesin still accumulates at CTCF sites but at lower levels compared to WT (Li et al, 2020). Likewise, cohesin processivity is affected in vitro by the large minichromosome maintenance (MCM) complex (Dequeker et al, 2022). This suggests that CTCF is sufficient to block cohesin without a specific interaction, and the same could be true for other large protein complexes. Therefore, there may be different methods to block loop extrusion; CTCF can directly interact with cohesin, causing it to be retained at CTCF-bound sites; however, passive blocking of cohesin by large multi-protein complexes may also occur and could provide a mechanism for how actively transcribed genes can behave as insulators/boundaries. This is also evident in cohesin ChIP-seq tracks as cohesin is highly enriched over CTCF sites, whilst enrichment over other active elements (including the insertion site in this manuscript) is less so. This concept of transcription interacting and/or interfering with cohesin processivity has been raised in previous studies (Hsieh et al, 2020; Zhang et al, 2023; Bonev et al,

2017). It is likely that formation of the α-globin subTAD is the result of a combination of mechanisms, both cohesin-dependent and independent, working together over development and differentiation to create a configuration for correct α-globin expression.

In summary, we provide evidence that, in addition to CTCF binding sites, actively transcribed genes may also behave as insulators and aid in forming the boundaries of TADs. This suggests that an active promoter may have multiple roles in shaping the genome.

## Methods

**Reagents and tools table**

| Reagent/resource | Reference or source | Identifier or catalogue number |
|---|---|---|
| **Experimental models** | | |
| Δθ1 Mouse, derived from C57BL/6J | This study | |
| Δθ2 Mouse, derived from C57BL/6J | This study | |
| Δθ1/2 Mouse, derived from C57BL/6J | This study | |
| ΔHS + 44/48 Mouse, derived from C57BL/6J | This study | |
| Δθ1/2/HS + 44/48 Mouse, derived from Δθ1/2 mouse | This study | |
| Hba-a1::mVenus hemizygous mESC | Francis et al, 2022 | |
| Ins[HS-38] Hba-a1::mVenus hemizygous mESC | Tsang et al, 2024 | |
| Ins[aP+G] Hba-a1::mVenus hemizygous mESC | This study | |
| Ins[aPonly] Hba-a1::mVenus hemizygous mESC | This study | |
| Ins[aGene] Hba-a1::mVenus hemizygous mESC | This study | |
| **Recombinant DNA** | | |
| pROSA-TV2 vector | Tsang et al, 2024 | |
| pROSA-TV2 vector, with inserts | This study, this paper, Twist Bioscience Synthesis of fragments and in-house cloning | |
| pCAGGS-Cre-IRESpuro plasmid | (Smith et al, 2002) provided by the Genome Engineering Facility at the Weatherall Institute of Molecular Medicine | |

| Reagent/resource | Reference or source | Identifier or catalogue number |
|---|---|---|
| **Antibodies** | | |
| Anti-Ter119-PE | Miltenyi Biotec | 130-102-336 |
| Anti-CD71-APC | eBioscience | 17-0711-80 |
| Anti-CD71-FITC | eBioscience | 11-0711-85 |
| Anti-Rad21 | Abcam | ab154769 (lot:11035529-12) |
| Anti-CTCF | Millipore | 07-729 (lot: 2836926) |
| **Oligonucleotides and other sequence-based reagents** | | |
| sgRNA Guide for Δθ1 | This study | 5′ TGGAACGATGCAGCGCCCCC |
| sgRNA Guide for Δθ2 | This study | 5′ TGAAACACAAGAGGCCGCCA |
| sgRNA Guide for Δθ2 | This study | 5′ GACATCTTTGAGCTCAGCCA |
| sgRNA Guide for ΔHS + 44 | This study | 5′ GAAAGCCAGTGGCGCCACCT |
| sgRNA Guide for ΔHS + 44 | This study | 5′CCCTGCAGGCCACTATAAGT |
| sgRNA Guide for ΔHS + 48 | This study | 5′ TCCAAGGTCCTCAAGCAGAC |
| sgRNA Guide for ΔHS + 48 | This study | 5′ CGACGAGCACCCCCGTGTGG |
| sgRNA Guide for insertions | Tsang et al, 2024 | 5′ GCTGTAGTGTAACTAACTGC |
| sgRNA Guide for insertions | Tsang et al, 2024 | 5′ GCTTCAAGAACTGCCTTCCTG |
| **Chemicals, enzymes and other reagents** | | |
| Anti-PE MACS microbeads | Miltenyi Biotec | 130-048-801 |
| Anti-FITC MACS microbeads | Miltenyi Biotec | 130-048-701 |
| MACS lineage selection columns | Miltenyi Biotec | 130-042-401 |
| Foetal bovine serum | Gibco | 10270106 |
| LIF | Bio-connect | GFM200-100 |
| Glasgow's MEM | Gibco | 21710082 |
| sodium pyruvate | Gibco | 11360-039 |
| L-gluatamine | Gibco | 25030-024 |
| MEM (NE) AA | Gibco | 11140-035 |
| beta-mercaptoethanol | Gibco | 31350-010 |
| IMDM with GlutaMAX | Gibco | 31980048 |
| Transferrin | Roche | 10652202001 |
| PFHM II | Gibco | 31980022 |
| L-Ascorbic Acid | Sigma | A-4544 |
| Monothioglycol | Sigma | M6145 |
| Pen/strep | Gibco | 15070 064 |
| 0.25% Trypsin EDTA | Gibco | 25200-056 |
| Lipofectamine™ LTX Reagent with PLUS™ Reagent | Thermofisher Scientific | cat# 15338100 |

| Reagent/resource | Reference or source | Identifier or catalogue number |
|---|---|---|
| 90 mm petri dish | ThermoFisher | 101VR20 |
| Chromatin Immunoprecipitation (ChIP) Assay Kit | Sigma | 17-295 |
| COmplete EDTA-free PIC | Roche | 11873580001 |
| Dynabeads Protein A | InVitrogen | 100-01D |
| Dynabeads Protein G | InVitrogen | 100-03D |
| ChIP Clean and Concentrate kit | Zymo | D5205 |
| NEBNext Ultra™ II DNA Library Prep Kit for Illumina | New England Biolabs | E7645S |
| Tagment DNA Enzyme Kit | Illumina | 20034211 |
| NEBNext High-Fidelity 2x PCR Master Mix | New England Biolabs | M0541 |
| NEBNext rRNA Depletion Kit (Human/Mouse/Rat) | New England Biolabs | E6310 |
| rRNA and globin mRNA using the Globin-Zero Gold rRNA Removal Kit | Illumina | GZG1224 |
| NEBNext Poly(A) mRNA Magnetic Isolation Module | New England Biolabs | E7490 |
| NEBNext Ultra II Directional RNA Library Prep Kit | New England Biolabs | E7760 |
| NEBNext Multiplex Oligos for Illumina | New England Biolabs | E7335/E7500 |
| Agencourt Ampure XP SPRI Beads | Beckman Coulter | A63881 |
| TRI reagent | Sigma-Aldrich | T9424 |
| Direct-zol RNA MiniPrep kit | Zymo Research | R2050 |
| SuperScript III First-Strand Synthesis SuperMix for qRT-PCR | Thermofisher | 11752-050 |
| TaqMan Universal PCR Master Mix | ThermoFisher | 4304437 |
| Hba-a1/2 Taqman probe | Thermofisher Scientific | Assay ID Mm00845395_s1 |
| Hbb-b1/2 Taqman probe | Thermofisher Scientific | Assay ID Mm01611268_g1 |
| Sh3pxd2b Taqman probe | Thermofisher Scientific | Assay ID Mm00616672_m1 |
| Ubtd2 Taqman probe | Thermofisher Scientific | Assay ID Mm00612868_m1 |
| Rn18s Taqman probe | Thermofisher Scientific | Assay ID Mm04277571_s1 |
| RPS18 Taqman probe | Thermofisher Scientific | Assay ID Mm02601777_g1 |

| Reagent/resource | Reference or source | Identifier or catalogue number |
|---|---|---|
| Tapestation D1000 screentape | Aligent | 5067-5583 |
| Tapestation RNA screentape | Aligent | 5067-5576 |
| DpnII | New England Biolabs | R0543M |
| T4 DNA Ligase, HC | Life Technologies Ltd | EL0013 |
| Dynabeads M-280 Streptavidin | ThermoFisher | 11205D |
| NimbleGen SeqCap EZ Hybridization and Wash Kit or | Roche | 05634261001 |
| NimbleGen SeqCap EZ Accessory Kit v2 | Roche | 07145594001 |
| HyperCapture Target Enrichment Kit | Roche | 9075828001 |
| **Software** | | |
| star (version 2.7.3a) | Dobin et al, 2013 | |
| Bowtie2 (version 2.4.2) | Langmead and Salzberg, 2012 | |
| Bowtie (version 1.2.3) | Langmead et al, 2009 | |
| Samtools (version 1.17) | Danecek et al, 2021 | |
| BCFtools (version 1.17) | Danecek et al, 2021 | |
| Bedtools (version 2.25.0) | Quinlan and Hall, 2010 | |
| Deeptools (version 3.5.2) | Ramírez et al, 2016 | |
| UpStreamPipeline | Riva et al, 2023 | |
| CCanalyser (CC2) | Davies et al, 2016 | |
| FLASH (version 1.2.11) | Magoc and Salzberg, 2011 | |
| Wiggletools (1.2.11) | Zerbino et al, 2014 | |
| Subread (version 2.0.6) | Liao, Smyth, and Shi, 2014 | |
| R (version 4.5.1) | RCoreTeam, 2025 | |
| DESeq2 (version 1.48.2) | Love, Huber, and Anders, 2014 | |
| BSgenomeForge (version 1.8.1) | Pagès and Kakopo, 2025 | |
| Plotgardener (version 1.14.0) | Kramer et al, 2022 | |
| FlowJo (version 10.9.0) | BD LifeSciences | |
| GraphPad Prism (version 10.6.1) | www.graphpad.com | |
| MEME Suite | Bailey et al, 2015 | |
| JASPAR | Ovek Baydar et al, 2025 | |
| **Other** | | |

| Reagent/resource | Reference or source | Identifier or catalogue number |
|---|---|---|
| NextSeq500 sequencer | Illumina | N/A |
| Bioruptor Pico sonicator | Diagenode | N/A |
| ME220 Focused-ultrasonicator | Covaris | N/A |
| Tapestation 2200 | Aligent | N/A |
| Attune NxT cytometer | ThermoFisher | N/A |
| QuantStudio 3 RT-PCR machine | Applied Biosystems | N/A |

## Animal procedure

The mutant and wild-type mouse strains reported in this study were generated and maintained on a C57BL/6J background in accordance with the European Union Directive 2010/63/EU and/or the UK Animal (Scientific Procedures) Act 1986, with procedures reviewed by the clinical medicine Animal Welfare and Ethical Review Body (AWERB). Experimental procedures were conducted under project licences PPL 30/3339 and PAA2AAE49. All animals were housed in Individually Ventilated Cages with enrichment, provided with food and water ad libitum, and maintained on a 12 h light: 12 h dark cycle (150–200 lux cool white LED light, measured at the cage floor). Mice were given neutral identifiers and analysed by research technicians unaware of the mouse genotype during outcome assessment. Sex of the animals has no impact on the molecular analyses in this study and was not included as a factor except for the haematological assessment. Samples for ATAC-seq, ChIP-seq, NG Capture-C, and gene expression analyses were not randomised, and the investigators were not blinded to allocation during these experiments and outcome assessments.

## Generation of mutant mouse strains

Mouse models harbouring mutations of CTCF binding sites around the mouse α-globin locus were generated using CRISPR-Cas9-mediated genome editing by either targeting mouse embryonic stem cells, which were then used in blastocyst injections, or by direct microinjection of zygotes. Preparation of CRISPR-Cas9 expression constructs for targeting of mouse embryonic stem cells and preparation of CRISPR-Cas9 reagents and ssODN templates, as required, for direct microinjection of zygotes were performed as previously described (Hanssen et al, 2017). For the generation of the $\Delta\theta1/\theta2/HS + 44/HS + 48$ model, oocytes/blastocysts from $\Delta\theta1–\theta2$ mice were microinjected with CRISPR/Cas9 expression constructs and sgRNA targeting directly upstream of HS + 44 and downstream of HS + 48, resulting in the deletion of both CBS and the ~4 kb intervening region. The 20-nucleotide guide sequences used to direct the Cas9 protein to the target CTCF binding sites, the ssODN donor sequences and genotyping PCR primer sequences are shown in Appendix Table S1.

## Generation of mouse embryonic stem cell genetic models

Mouse embryonic stem cells (mESCs, derived from E14TG2a) were maintained using standard protocols, on gelatinised plates with complete supplemented GMEM media with FBS and LIF, as described previously (Tsang et al, 2024). The *Hba-a1::mVenus* hemizygous mESC line used for the generation of the assay models, the CRISPR-Cas9 targeting and PCR genotyping strategies for inserting sequences of interest into the landing site (chr11: 32171787-32171856 mm9) were described in detail previously (Tsang et al, 2024; Francis et al, 2022) (Appendix Fig. S2). The α-globin sequences inserted ectopically into the α-globin locus in this study and the sgRNA sequences used for model generation are shown in Appendix Table S2.

## Isolation of erythroid cells derived from adult mouse spleen

Primary Ter119+ erythroid cells were obtained from the spleens of adult mice that were treated with phenylhydrazine as described previously (Spivak et al, 1973). Spleens were mechanically dissociated into single-cell suspensions in cold phosphate-buffered saline (PBS; Gibco: 10010023)/10% foetal bovine serum (FBS; Gibco: 10270106) and passed through a 70-μm filter to remove clumps. Cells were washed with cold PBS/10% FBS and resuspended in 10 μl of cold PBS/10% FBS per $10^6$ cells and stained with a 1/100 dilution of anti-Ter119-PE antibody (Miltenyi Biotec: 130-102-336) at 4 °C for 20 min. Stained cells were washed with cold PBS/10% FBS and resuspended in 8 μl of cold PBS/0.5% BSA/ 2 mM EDTA and 2 μl of anti-PE MACS microbeads (Miltenyi Biotec: 130-048-801) per $10^6$ cells and incubated at 4 °C for 15 min. Ter119+ cells were positively selected via MACS lineage selection columns (Miltenyi Biotec: 130-042-401) and processed for downstream applications. Purity of the isolated erythroid cells was routinely verified by Fluorescence-Activated Cell Sorting (FACS). FACS data were analysed with FlowJo (version 10.9.0 BD LifeSciences).

## Isolation of CD71+ erythroid cells derived from in vitro embryoid body differentiation

Genetically modified mESCs were differentiated in vitro as described previously (Francis et al, 2022). CD71+ cells derived from this protocol closely resemble primitive erythroblasts. Base media for differentiation was composed of IMDM + 1× Glutamax, 1× Pen/Strep and 140 μM MTG. Gibco FCS was heat-inactivated (ΔFCS) at 56 °C for 1 h. mESCs were grown for 48 h in Adaptation media (15% ΔFCS, 1000 U/ml LIF in Base media). Cells were then de-adhered using 0.05% Trypsin and resuspended to a single-cell suspension in Resuspension media (10% ΔFCS in Base media, no LIF). Cells were seeded into Differentiation media (Base media supplemented with 15% ΔFCS, 5% PFHM II, 1× L-Glutamine, 1% Transferrin (Merck 10652202001), 1% Ascorbic Acid and additional MTG). Seeding density was between 10,000 to 20,000 cells per 10 cm plate, in 10–15 ml differentiation media. Plates were incubated at 37 °C (5% $CO_2$) for a total of 7 days with daily shaking to agitate the cells to discourage cell adherence. The resulting Embryoid bodies (EBs) were collected, washed once in PBS then trypsinised in 0.25% Trypsin at 37 °C for 5 min with shaking until fully disaggregated. Cells were quenched with 10% FCS in PBS and strained through a 70-μm cell strainer. For determining YFP fluorescence from reporter cells, a small aliquot of cells was taken prior to column selection and stained with APC-conjugated α-CD71 antibody (1:8000, eBioscience: 17-0711-80) and Hoechst (1:10,000) at 4 °C for 30 min, then washed in FACS buffer and analysed on an Attune NxT cytometer. The gating strategy comprised of gating for live single cells then stringent gating was used for CD71 + /YFP+ cells; median YFP florescence was measured in FlowJo (version 10.9.0 BD LifeSciences) then plotted and statistics performed in GraphPad Prism (for MacOS version 10.6.1 (www.graphpad.com)).

To isolate CD71+ cells, total disaggregated EB material was resuspended in 3-5 ml FACS buffer with 2 μl FITC-conjugated α-CD71 antibody (eBioscience: R17217, 11-0711-85) per $1 \times 10^7$ cells. Cells were left to roll at 4 °C for 20 min, quenched with FACS buffer, then resuspended in 200 μl PBE (2 mM EDTA pH 8, 0.5% BSA in PBS) with 20 μl α-FITC beads (Miltenyi Biotec:130-048-701) per $1 \times 10^7$ cells and left to roll at 4 °C for 15 min. Stained cells were washed, then resuspended in cold PBE, then loaded via a 30-μm strainer into an equilibrated LS column (Miltenyi Biotec: 130-042-401) on a MACS magnet. Magnetised columns were washed three times with cold PBE (negative fraction). The CD71-positive fraction was collected by releasing the column from the magnet, adding PBE and applying pressure with a plunger. The cells were resuspended in FACS buffer for counting and preparation for other assays.

## Custom genome generation

Custom genome references were generated for analysis. For *Hba-a1/a2* SNP-specific counts, a custom genome was created based on mm39 with only one copy of *Hba* (*Hba-a1*) by deleting the *Hba-a2* sequence from the genome fasta, and the informative SNP (mm39: chr11:32234343) masked using the bedtools maskfasta function (Bedtools version 2.25.0) (Quinlan and Hall, 2010). The masked fasta was used to create a STAR index (--runMode genomeGenerate) used to align fastqs to the custom genome (STAR version 2.7.3a) (Dobin et al, 2013). This allowed all reads to map to one reference sequence allowing determination of their source by using unique SNPs to each variant.

For alignments in the landing pad model, a custom genome was made based on mm9 with the native *Hba-a1* replaced with *Hba-a1*:2Ap:*mVenus*:SV40NLS sequence. Positions of expected SNPs were masked and genome indexes generated for STAR (version 2.7.3a) (--runMode genomeGenerate) (Dobin et al, 2013) and Bowtie2 (version 2.4.2) (bowtie2-build) alignments (Langmead and Salzberg, 2012). For visualising, BSgenomeForge (version 1.8.1) (Pagès and Kakopo, 2025) was used to convert the custom genome for use in R and plots were generated with Plotgardener (version 1.14.0) (Kramer et al, 2022).

## ATAC-seq

ATAC-seq was performed on 75,000 Ter119+ cells isolated from phenylhydrazine-treated mouse spleens as previously described (Buenrostro et al, 2013) or CD71+ cells from in vitro differentiation of mESCs (Francis et al, 2022). ATAC-seq libraries were sequenced on the Illumina Nextseq platform using a 75-cycle

paired-end kit (NextSeq 500/550 High Output Kit v2.5: 20024906). Data were analysed using an in-house pipeline using Bowtie2 (Langmead and Salzberg, 2012) with multimapping active (-k 2) to map reads to the mm9 or to a custom genome reference. Samtools (version 1.17) was used to filter for mapped reads, sort and remove PCR duplicates (Danecek et al, 2021), and replicates were normalised to Reads Per Kilobase per Million (RPKM) mapped reads using deeptools (version 3.5.2) bamCoverage (Ramírez et al, 2016). For visualisation, ATAC-seq data were averaged across replicates using wiggletools (version 1.2.11) (Zerbino et al, 2014) and plotted with Plotgardener (version 1.14.0) (Kramer et al, 2022).

## ChIP-seq

CTCF motif analysis was performed with FIMO (MEME-suite) motif analysis (Bailey et al, 2015) and mouse JASPAR position weight matrices (Ovek Baydar et al, 2025). CTCF ChIP-seq was performed on $1 \times 10^7$ Ter119+ erythroid cells using a ChIP Assay Kit (Millipore: 17-295) according to the manufacturer's instructions. Cells were crosslinked by a single 10 min 1% formaldehyde fixation. Chromatin fragmentation was performed with the Bioruptor Pico sonicator (Diagenode) for a total sonication time of 4 min (8 cycles) at 4 °C to obtain an average fragment size between 200 and 400 bp. Immunoprecipitation was performed overnight at 4 °C with an anti-CTCF antibody (Millipore: 10 µl 07-729, lot: 2836926). Rad21 ChIP-seq was performed on 1-3 ×10⁶ CD71+ cells as described in (Georgiades et al, 2025). Sonicated chromatin (200-500 bp) was prepared using a Covaris ME220 with the following settings: 600secs, 75 power, 25% Duty Factor, 1000 cycles per burst and precipitated using anti-Rad21 antibody (Abcam: 10 µl ab154769, lot: 11035529-12). Library preparation of immunoprecipitated DNA fragments was performed using NEBNext Ultra II DNA Library Prep Kit for Illumina (E7645) according to the manufacturer's instructions. Libraries were sequenced on the Illumina Nextseq platform using a 75-cycle paired-end kit (NextSeq 500/550 High Output Kit v2.5: 20024906). Data were analysed using an in-house pipeline using Bowtie2 (Langmead and Salzberg, 2012), with multimapping active (-k 2), to map reads to mm9 or a custom genome reference. Samtools (version 1.17) was used to filter for mapped reads, sort and remove PCR duplicates (Danecek et al, 2021), and replicates were normalised (RPKM) using deeptools (version 3.5.2) bamCoverage (Ramírez et al, 2016). For visualisation, unless specified otherwise, ChIP-seq data were averaged across replicates using wiggletools (version 1.2.11) (Zerbino et al, 2014) and plotted with Plotgardener (version 1.14.0) (Kramer et al, 2022). Region counts were generated by using the deeptools MultibigwigSummary package from RPKM normalised bigwigs (binsize=1). These counts were normalised to a region within the Beta-globin locus (mm9, chr7:110983901-111027888), which would be expected to have consistent read coverage between models and hence was used to normalise the counts across samples.

## NG Capture-C

Next-generation Capture-C was performed as previously described (Davies et al, 2016). A total of $1-2 \times 10^7$ Ter119+ erythroid cells were used per biological replicate or a total of $1-3 \times 10^6$ in vitro-derived CD71+ erythroid cells per replicate. On the day of harvest,

cells were fixed with fixed with 2% formaldehyde for 10 min, before quenching with 125 mM Glycine. Cells were spun down, washed with PBS, and the pellet suspended in a mild NP-40-containing lysis buffer. Samples were then snap frozen and stored at −80 °C. 3C libraries were prepared using the DpnII-restriction enzyme for digestion. Illumina TruSeq adaptors were added using the NEBNext Ultra II DNA Library Prep Kit for Illumina (E7645) according to the manufacturer's instructions, and performed capture enrichment using NimbleGen SeqCap EZ Hybridization and Wash Kit (Roche: 05634261001), NimbleGen SeqCap EZ Accessory Kit v2 (Roche: 07145594001) or HyperCapture Target Enrichment Kit (Roche: 9075828001), and previously published custom biotinylated DNA oligonucleotides (R1, R2 and HS-38 viewpoints (Hanssen et al, 2017); α-globin promoters, mitoferrin, and β-blobin viewpoints (Davies et al, 2016). The data were then analysed and visualised using a Capture-C pipeline within an in-house copy of the UpStreamPipeline suite (Riva et al, 2023), which incorporates the original Capture-C analyser scripts (Davies et al, 2016): briefly, adaptors are removed using Trim-galore (version 0.6.10); read pairs reconstructed to single reads using FLASH (version 1.2.11) (Magoc and Salzberg, 2011); reads are in silico digested using DpnII recognition sites; aligned to reference genomes with Bowtie (Langmead et al, 2009); then viewpoint/reporter counts assigned to DpnII fragments genome wide, and PCR duplicates removed (Davies et al, 2016). This was performed on mm9 or a custom genome reference. *Cis* reporter counts for each sample were normalised to the highest count across all samples, then the mean and standard deviation were calculated across replicates for each genotype. Counts were smoothed for visualisation using a one-dimensional Gaussian filter with sigma set to 0.75. Non-normalised interaction counts were assessed for significant changes between samples by DESeq2 (version 1.48.2) (Love et al, 2014) in R (version 4.5.1) (RCoreTeam, 2025). Differential tracks were generated by first normalising in *cis* to a total of 100,000 reporters, with counts mapping near to the viewpoints excluded, then the mean between replicates was calculated, and the averaged counts of one condition were subtracted from the other. The region highlighted upstream of *Hba-a1* was chosen such that reporters should be specific to the native gene and not the inserted variant; note in both smoothed and differential profiles, counts contributing to *Hba-a1/2* originate from both native and inserted variants.

## RNA expression analysis

Total RNA was isolated from $5 \times 10^6$ Ter119+ erythroid cells lysed in TRI reagent (Sigma-Aldrich: T9424) using a Direct-zol RNA MiniPrep kit (Zymo Research: R2050). DNase I treatment was performed on the column as recommended in the manufacturer's instructions, but with an increased incubation of 30 min at room temperature (rather than 15 min). To assess relative changes in gene expression by RT-qPCR, cDNA was synthesised from 1 µg of total RNA using SuperScript III First-Strand Synthesis SuperMix for qRT-PCR (Invitrogen, ThermoFisher: 11752-050) according to the manufacturer's instructions. The ΔΔCt method was used for relative quantification of RNA abundance using TaqMan Universal PCR Master Mix (Applied Biosystems, ThermoFisher: 4304437) and the following TaqMan probes: Mm00845395_s1 (*Hba-a1/2*), Mm01611268_g1 (*Hbb-b1*), Mm00616672_m1 (*Sh3pxd2b*), Mm00612868_m1 (*Ubtd2*), and Mm04277571_s1 (*Rn18s*). The

ΔΔCt method was used to quantify the relative RNA abundances. For α:β ratios, the second ΔΔCt normalisation was performed against β-globin (Hba ΔCT/Hbb ΔCT).

For RNA-seq libraries from Ter119+ erythroid cells, 1–2 μg of total RNA was depleted of rRNA and globin mRNA using the Globin-Zero Gold rRNA Removal Kit (Illumina: GZG1224) according to the manufacturer's instructions. For RNA-seq libraries from CD71+ in vitro-derived erythrocytes, 500 ng of total RNA was depleted of rRNA only using NEBNext rRNA depletion kit (New England Biolabs: E6310). To enrich for mRNA, poly(A) + RNA was isolated, strand-specific cDNA was synthesised, and the resulting libraries were prepared for Illumina sequencing using the NEBNext Poly(A) mRNA Magnetic Isolation Module (New England Biolabs: E7490) and the NEBNext Ultra II Directional RNA Library Prep Kit for Illumina (New England Biolabs: E7760) following the manufacturer's instructions. Poly(A) + RNA-seq libraries were sequenced on the Illumina Nextseq platform (NextSeq 500/550 High Output Kit v2.5: 20024906). Reads were aligned to a custom mm39 genome reference with SNPs masked (see above) using STAR (version 2.7.3a) (Dobin et al, 2013), samtools (version 1.17) was used to filter for mapped reads, sort and remove PCR duplicates (Danecek et al, 2021). To perform variant calling analysis on RNA-seq reads originating from α-globin transcripts, counts over SNPs were performed using BCFtools (version 1.17) mpileup (--min-BQ 30). Counts of each base were then expressed as a fraction of the total count per replicate; to determine if there were differences between samples, a one-way ANOVA with a Tukey post hoc test was performed on the *Hba-a1* proportion between genotypes. To produce normalised counts and determine significance of changes in expression, subread FeatureCounts (version 2.0.6) (Liao et al, 2014) was performed using RefSeq gene annotation with custom annotations added as appropriate (--countReadPairs -M -O --ignoreDup) and differential gene analysis was performed with DESeq2 (version 1.48.2) (Love et al, 2014) in R (version 4.5.1)(RCoreTeam, 2025), and used to generate normalise counts and LFC shrinking (apeglm) performed. Directional rpkm normalised bigwigs were generated using deeptools bamCoverage (--filteredRNAstrand) (Ramírez et al, 2016).

## Statistical analysis

One-way ANOVA with Tukey multiple comparisons of means with 95% family-wise confidence level was applied to all RT-qPCR and FACS summary data using GraphPad Prism (version 10.6.1) (www.graphpad.com). Two-way ANOVA was performed for haematology measures, on a sample size greater than the minimum to detect an anaemic phenotype ($n \geq 3$). Chi-squared Goodness of Fit test was performed for assessing Mendelian ratios of mice. The statistical details of experiments can be found in the figure legends and depicted graphically in figures, including the statistical tests used, exact value of n, what n represents (e.g., number of animals, number of samples used). For the bioinformatic analyses, statistical solutions for differential detection of reads were performed with Bioconductor DESeq2 (version 1.48.2) (Love et al, 2014) in R (version 4.5.1)(RCoreTeam, 2025), significance was determined as padj<0.01, where padj is a Benjamini–Hochberg corrected Wald test *P* value.

## Data availability

All ATAC-seq, ChIP-seq, RNA-seq and Capture-C data generated for this study have been submitted to the NCBI Gene Expression Omnibus under accession numbers GSE153209 and GSE313947. Previously published data referenced in this work can be found under GSE97871 and GSE27921 (ChIP-seq in primary erythroid cells) and GSE137477 (Tiled-C in primary erythroid cells). All other data supporting the findings of this study are available from the corresponding author on reasonable request.

The source data of this paper are collected in the following database record: biostudies:S-SCDT-10_1038-S44318-026-00730-2.

## Peer review information

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

## Acknowledgements

The authors would like to express their gratitude to all who contributed to this work; we thank Simone Riva for access to the Capture-C pipeline in the CATCH-UP/UpstreamPipeline suite. This work was supported by the Wellcome Trust, Genomic Medicine and Statistics D.Phil Programme for CLH reference 109110/Z/15/Z; Chromosome and Developmental Biology D.Phil Programme, for LJC and LLPH, and RS, references 222843/Z/21/Z, 099684/Z/12/Z and 215111/Z/18/Z; Wellcome Trust Strategic Award, for JRH reference 106130/Z/14/Z; Wellcome Trust Core Award, for BD, reference 203141/Z/16/Z, The UKRI - Medical Research Council for DRH, MRC Core Funding and Project Grant, reference MR/N00969X/1 and Programme Grant MR/T014067/1. DRH is also supported by The Chinese Academy of Medical Sciences (CAMS) Innovation Fund for Medical Science (CIFMS) (grant: 2018-I2M-2-002). MTK is also funded by BBSRC-NSF/BIO (BB/Y008898/1). This work was also supported by the Genome Engineering and Transgenics facility, the Flow Cytometry facility, CCB: High-Performance Computing facility and Sequencing facilities at the Weatherall Institute of Molecular Medicine.

## Author contributions

**Lucy J Cornell**: Data curation; Formal analysis; Investigation; Visualisation; Writing—original draft; Writing—review and editing. **Caroline L Harrold**: Data curation; Formal analysis; Investigation; Visualisation; Writing—original draft. **Susannah Holliman**: Data curation; Formal analysis; Investigation; Visualisation; Writing—review and editing. **Felice Tsang**: Data curation; Investigation. **Matthew E Gosden**: Data curation; Formal analysis; Investigation. **Lars L P Hanssen**: Conceptualisation; Investigation. **Rosa Stolper**: Data curation; Formal analysis; Investigation. **Damien J Downes**: Formal analysis; Visualisation. **Daniel Biggs**: Resources; Methodology. **Chris Preece**: Resources; Methodology. **Samy Alghadban**: Resources; Methodology. **Jacqueline A Sharpe**: Resources; Methodology. **Benjamin Davies**: Resources; Methodology. **Jackie A Sloane-Stanley**: Resources; Methodology. **Jim R Hughes**: Conceptualisation; Supervision; Methodology. **Douglas R Higgs**: Conceptualisation; Supervision; Methodology; Writing—review and editing. **Mira T Kassouf**: Conceptualisation; Supervision; Funding acquisition; Investigation; Methodology; Writing—original draft; Writing—review and editing.

Source data underlying figure panels in this paper may have individual authorship assigned. Where available, figure panel/source data authorship is listed in the following database record: biostudies:S-SCDT-10_1038-S44318-026-00730-2.

## Disclosure and competing interests statement

JRH is a founder and shareholder of Nucleome Therapeutics; JRH and DJD are paid consultants for Nucleome Therapeutics. JRH holds patents for NG Capture-C. Patent applicant: Oxford University Innovation Limited, Name of inventor(s): James R Hughes and James Davies nos. WO2017068379A1, EP3365464B1 and US10934578B2. Specific aspect of manuscript covered in patent application: NG-CaptureC experiments. These authors declare no other financial or non-financial interests. The remaining authors declare no competing interests.

# Expanded View Figures

**Figure EV1. 3D structure and cis-regulatory elements of the α globin locus.**

An expanded view of the α-globin locus in primary Ter119+ erythroid cells. Top panel shows Tiled-C interaction heatmap at 2 kb resolution adapted from (Oudelaar et al, 2020), the horizontal grey bars between the tracks represent the ~70 kb α globin sub-TAD (light grey, mm9 chr11:32,136,000–32,202,000) nested within a larger ~165 kb TAD (dark grey, mm9 chr11:32,080,000–32,245,000), these domains are further highlighted as triangles in the Tiled-C panel. The adult α globin genes are highlighted in red. The individual alpha globin superenhancer elements (R1, R2, R3, Rm & R4) are highlighted in grey. Below show chromatin characterisation of the region in primary definitive erythroid cells; ATAC-seq (black, this study), H3K27ac ChIP-seq (light green) (Kowalczyk et al, 2012), PolII (dark green) Rad21 (blue) and CTCF ChIP-seq (dark blue) (Hanssen et al, 2017). The orientation of CTCF motifs is shown under peaks by red (forward) and blue (reverse) arrows. Bottom panels show ATAC, CTCF ChIP-seq (labelled with CTCF site identifiers) and Tiled-C interactions zoomed in on the TAD region (mm9 chr11:32,050,000–32,250,000).

▶

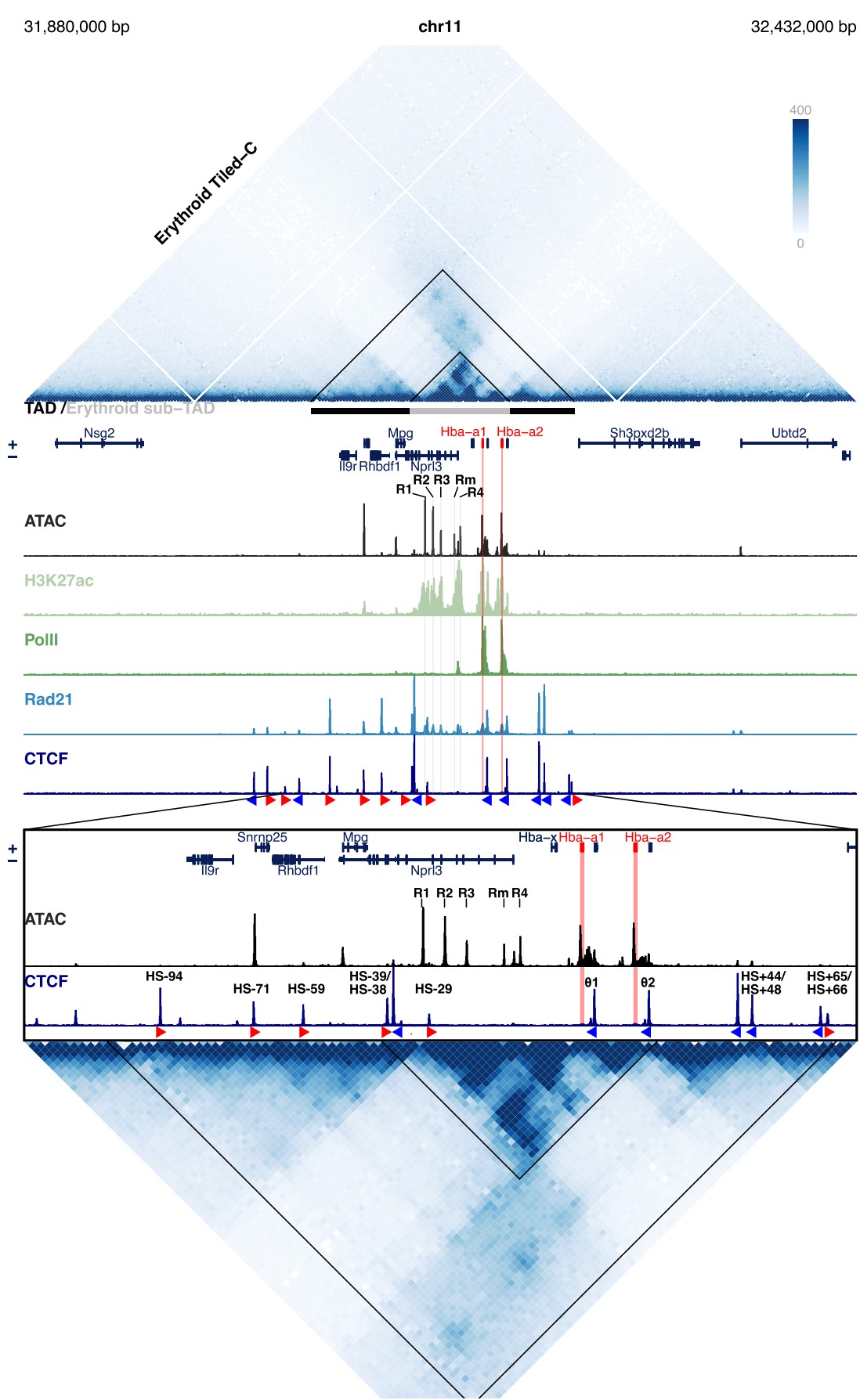

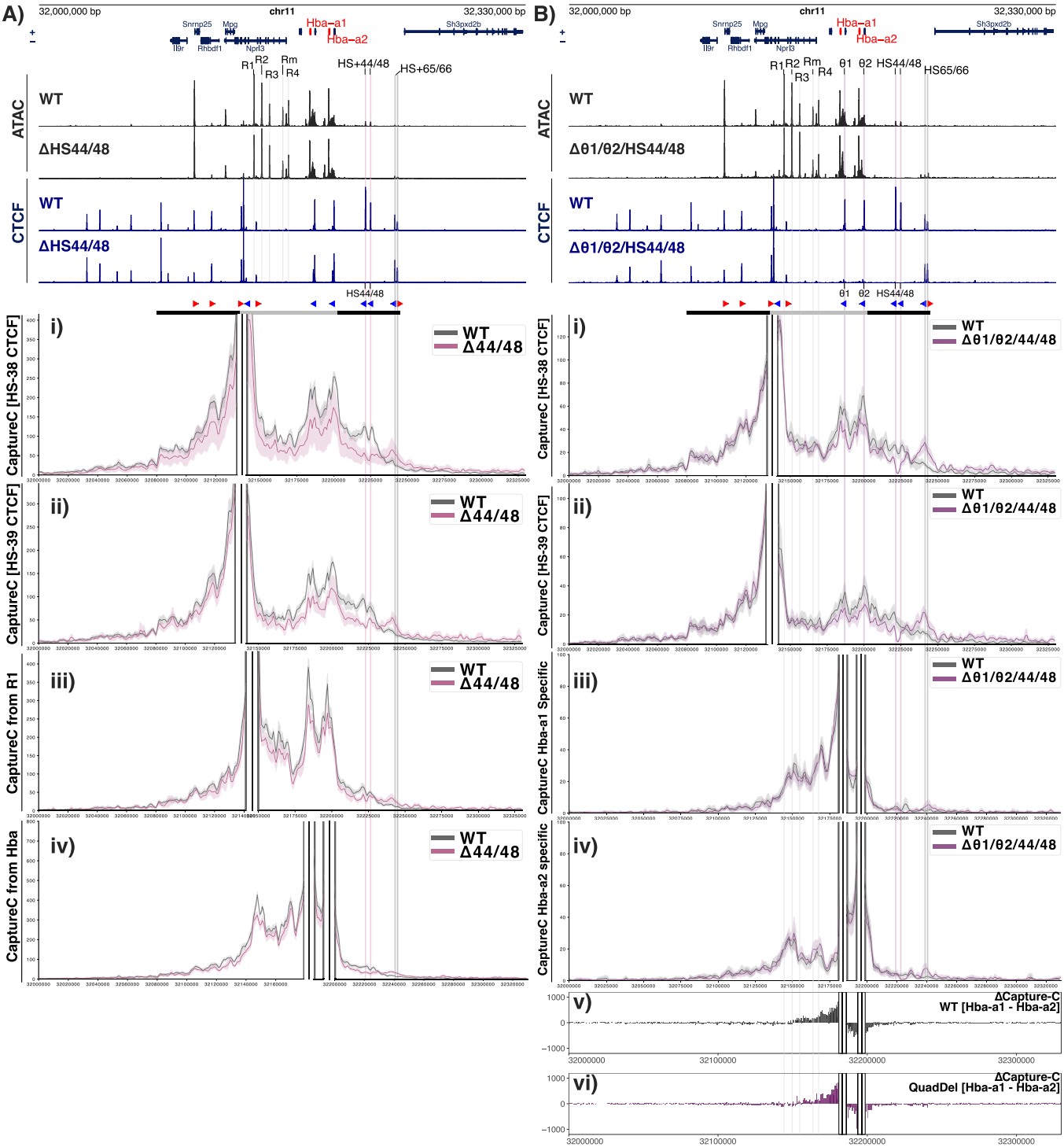

◀ **Figure EV2.  Capture-C interaction profiles of the α globin locus from various viewpoints in Δ44/48 and Δθ1/θ2/44/48 erythroid cells.**

(A) Top tracks show profiles for ATAC-seq (black) and CTCF ChIP-seq (navy) in primary erythroid cells (Ter119 + ) isolated from WT and ΔHS44/48 mice. Profiles show normalised (RPKM) and averaged data from $n = 3$ biological replicates across the α-globin locus (mm9, chr11:32,000,000–32,330,000) with genes and genomic position, with positioning of genes above or below representing sense and antisense transcription, respectively. The adult α globin genes are highlighted in red. The individual α globin superenhancer elements (R1, R2, R3, Rm, and R4) are highlighted in grey. The horizontal grey bars between the tracks represent the ~70 kb α globin sub-TAD (light grey, chr11:32,136,000–32,202,000) nested within a larger ~165 kb TAD (dark grey, chr11:32,080,000–32,245,000). The orientation of CTCF motifs is shown under peaks by red (forward) and blue (reverse) arrows. NG Capture-C interaction profiles of the α-globin locus from WT (grey) and ΔHS44/HS48 (purple) Ter119+ primary erythroid cells, the following viewpoints: (i) HS-38 CTCF, (ii) HS-39 CTCF, (iii) R1 enhancer element and (iv) Hba-a1/2 genes. The profiles represent normalised and averaged unique interactions from $n = 3$ biological replicates, smoothed with a 1D Gaussian filter. (B) As in (A) but showing profiles from primary APH-treated spleen cells isolated from WT and Δθ1/θ2/HS + 44/HS + 48 mice. NG Capture-C interaction profiles of the α-globin locus from WT (grey) and Δθ1/θ2/HS + 44/HS + 48 (purple) from the following viewpoints: (i) HS-38 CTCF, (ii) HS-39 CTCF, (iii) Hba-a1 SNP-specific interactions and (iv) Hba-a2 SNP-specific. The profiles represent normalised and averaged unique interactions from $n = 3$ biological replicates and halos representing ± standard deviation, smoothed with a 1D Gaussian filter. Differential tracks (ΔCaptureC) show subtractions (v) WT [Hba-a1 – Hba-a2] and (vi) Δθ1/θ2/HS + 44/HS + 48 [Hba-a1 – Hba-a2]) of the mean number of unique interactions per restriction fragment, scaled to a total of 100,000 interactions in cis from SNP-specific counts.

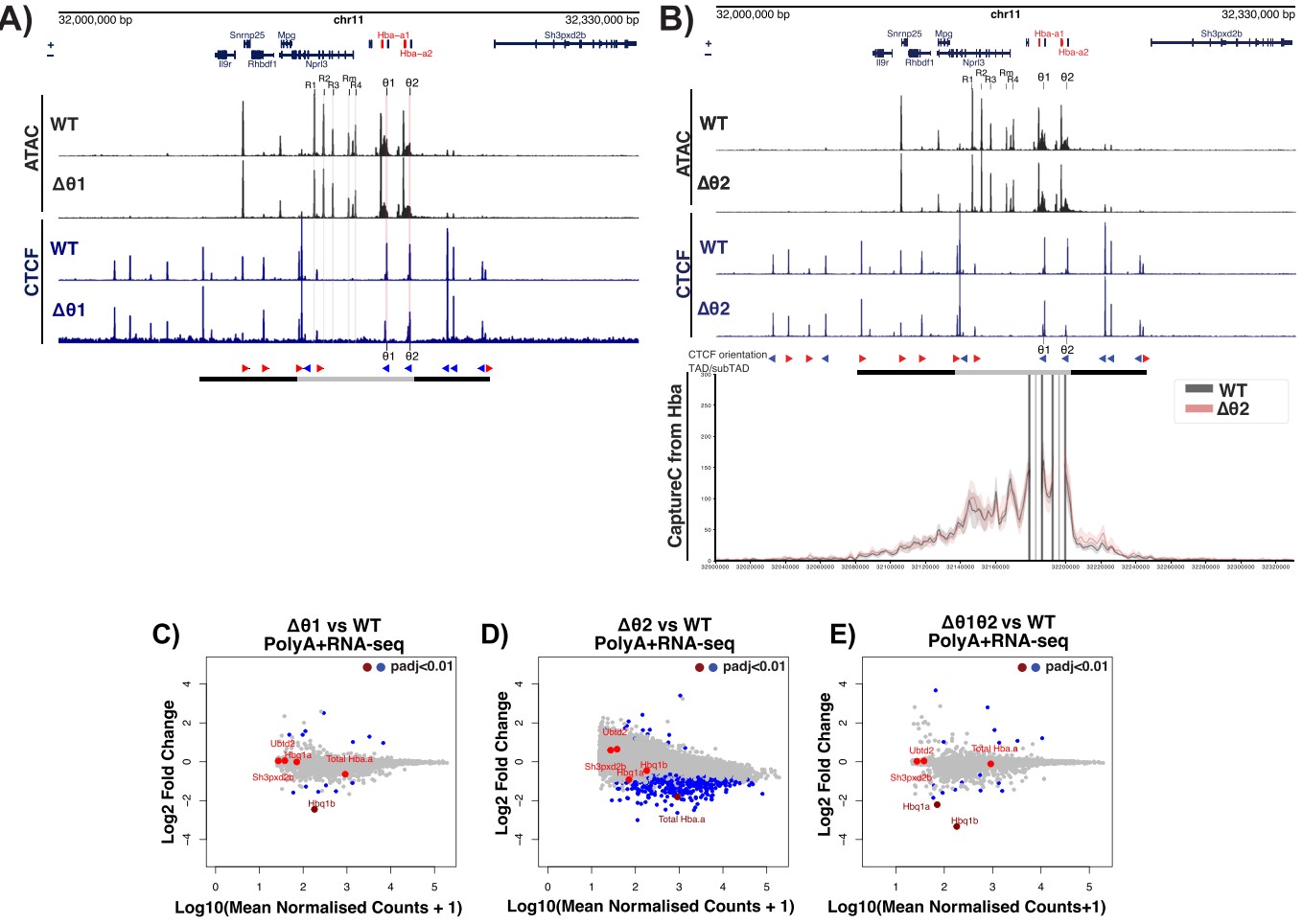

**Figure EV3. Chromatin accessibility, CTCF binding and PolyA+RNA-seq in Δθ1 and Δθ2 erythroid cells.**

(A) Characterisation of Δθ1 primary erythroid cells. Top tracks show profiles for ATAC-seq (black) and CTCF ChIP-seq (navy) in primary erythroid cells (Ter119+) isolated from WT and Δθ1 mice. Profiles show normalised (RPKM) and averaged data from $n = 3$ biological replicates across the α-globin locus (annotations are as in Fig. 1A). (B) Characterisation of Δθ2 primary erythroid cells. Top tracks show profiles for ATAC-seq (black) and CTCF ChIP-seq (navy) in primary erythroid cells (Ter119+) isolated from WT and Δθ2 mice. Profiles show normalised (RPKM) and averaged data from $n = 3$ biological replicates across the α-globin locus (annotations are as in Fig. 1A). NG Capture-C interaction profiles of the α-globin locus from the combined viewpoint of the *Hba* genes each with an exclusion zone, in WT (grey) and Δθ1/θ2 (pink) Ter119+ primary erythroid cells. The interaction profiles represent normalised and averaged unique interactions from $n = 2$ biological replicates and halos representing ± standard deviation, smoothed with a 1D Gaussian filter. (C–E) Differential expression (PolyA + RNA-seq) in Δθ1, Δθ2 and Δθ1/θ2 primary erythroid cells. MAplot of Log2 Fold change versus Log10 of normalised counts in the models above vs WT; each dot represents a gene. Genes with an adjusted *P* value (padj, Benjamini–Hochberg corrected) <0.01 are highlighted in blue. Genes of interest are highlighted in red (3′genes *Sh3pxd2b*, *Ubtd2*, Total Hba (*Hba-a1/2*) and θ1/θ2 associated genes *Hbq1b/a* respectively) and those with a significant difference from WT highlighted with dark red. Results from $n = 3$ biological replicates from each genotype. There is an unexpectedly high number of differential genes in the Δθ2 model further *Hbq1a* appears unchanged upon Δθ2, which is incongruent with the result in Δθ1/θ2. As the Δθ1/θ2 does not show these differences and is a combinatorial deletion of both θ1/θ2 and we must assume these differences in Δθ2 expression are due to technical error.

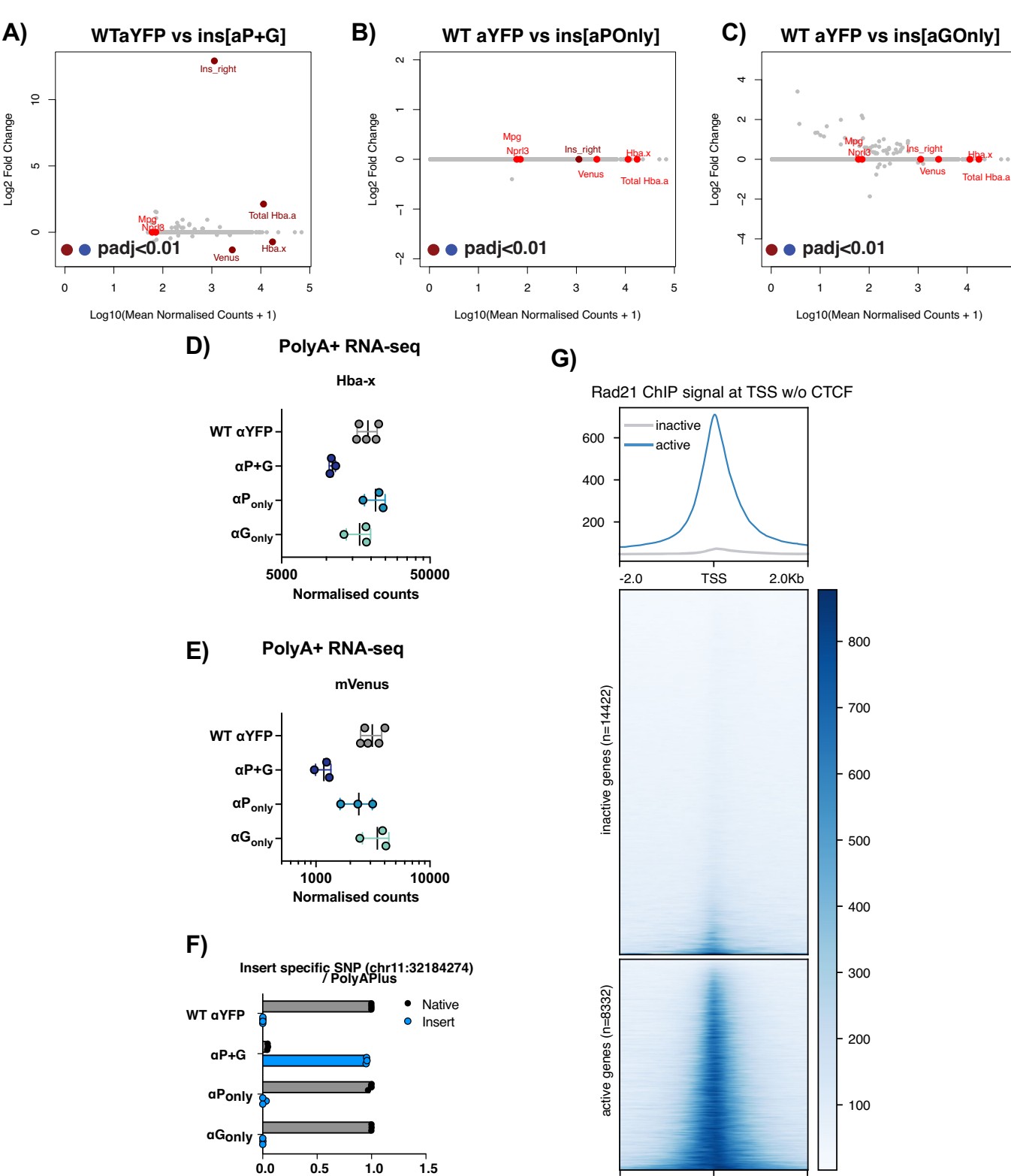

**Figure EV4. Extended characterisation of in vitro-derived CD71+ cells with inserts.**

(A–C) Differential expression between the WT aYFP reporter and the insertion models. MAplot of Log2 fold change in gene expression relative to WT against the Log10 of read counts; each dot represents a gene. Highlighted in red are Hba-a (representing total *Hba-a1/2* and aP+G in the P + G model), the region downstream of the insertion site (ins_right) and other genes in the locus. Significant changes plotted with dark red or blue (Wald test *P* value, Benjamini–Hochberg corrected: *P*adj<0.01), non-significant changes are in grey and bright red. Data is n = 5 replicates of WT-aYFP and n = 3 replicates of each other genotype. As these libraries were not globin-depleted, the representation is skewed toward globin genes. (D) *Hba-x* expression as normalised counts in PolyA-Plus RNA-seq. Data is representative of n = 5 replicates of WT-aYFP and n = 3 replicates of each other genotype. Error bars represent ± standard deviation. (E) *mVenus* expression as normalised counts in PolyA-Plus RNA-seq. Data is representative of n = 5 replicates of WT-aYFP and n = 3 replicates of each other genotype. Error bars represent ± standard deviation. (F) Exonic SNP-specific count PolyA-Plus RNA-seq. The aP+G gene had its own unique SNP in exon 3 allowing counting of the proportion of transcripts between the inserted or native α-globin copies. SNPs specific counts of each variant of *Hba* were counted similarly to Fig. 1E in RNA-seq from the CD71+ cells. Error bars represent ± standard deviation. (G) Heatmap and summary profile displaying Rad21 ChIP-seq signal in WT in vitro-derived CD71+ erythroid cells at non-redundant transcription start sites (TSS) of inactive (14422) and active (8332) genes, which do not have a CTCF binding site within a 2 kb window around the TSS. Source data are available online for this figure.

