## [Peer Review File · The EMBO Journal]

A functional overlap between actively transcribed genes and chromatin insulator elements

Lucy Cornell, Caroline Harrold, Susannah Holliman, Felice Tsang, Matthew Gosden, Lars Hanssen, Rosa Stolper, Damien Downes, Daniel Biggs, Chris Preece, Samy Alghadban, Jacqueline Sharpe, Benjamin Davies, Jackie Sloane-Stanley, Jim Hughes, Douglas Higgs, and Mira Kassouf

Corresponding author(s): Mira Kassouf (mira.kassouf@imm.ox.ac.uk) , Douglas Higgs (doug.higgs@imm.ox.ac.uk), Jim Hughes (jim.hughes@imm.ox.ac.uk)

Review Timeline:

Submission Date:	22nd Oct 24
Editorial Decision:	3rd Dec 24
Revision Received:	13th Oct 25
Editorial Decision:	25th Nov 25
Revision Received:	23rd Dec 25
Accepted:	21st Jan 26

Editor: Cornelius Schneider

Transaction Report:

Dear Dr. Kassouf,

Thank you for submitting your manuscript for consideration by the EMBO Journal. It has now been seen by three referees whose comments are shown below.

Given the referees' positive recommendations, I would like to invite you to submit a revised version of the manuscript, addressing the comments of all three reviewers. I am happy to discuss any questions regarding the revisions by email or videoconferencing.

I should add that it is EMBO Journal policy to allow only a single round of revision, and acceptance of your manuscript will therefore depend on the completeness of your responses in this revised version.

Thank you for the opportunity to consider your work for publication. I look forward to your revision.

Yours sincerely,

Cornelius Schneider, PhD
Editor
The EMBO Journal
c.schneider@embojournal.org

We realize that it is difficult to revise to a specific deadline. In the interest of protecting the conceptual advance provided by the work, we recommend a revision within 3 months (3rd Mar 2025). Please discuss the revision progress ahead of this time with the editor if you require more time to complete the revisions. Use the link below to submit your revision:

Referee #1:

In this manuscript, Cornell et al. investigate whether transcriptionally active genes can act as boundaries of 3D chromatin structures. Their study centers on the alpha globin gene which lies within a CTCF flanked TAD in mouse erythroid cells, where it is actively transcribed. They find deletion of the downstream CTCF motifs do not affect surrounding chromatin contacts or gene expression, suggesting the gene itself may define the boundary. In a previously validated insulation assay, they show the full-length alpha globin gene can act as an insulator, with the promoter or gene body alone showing modest insulator capabilities. This appears to be cohesin dependent, as it accumulates at the gene as we would expect it to at a CTCF boundary. Overall, the authors' careful dissection of this locus offers a new way to define boundaries independent of insulator proteins, opening the door to future studies into how widespread this phenomenon is.

I believe the paper would benefit from the following:

Major comments:

- It is unclear whether the insulation potential of the alpha globin gene is unique or if the same would be seen for any highly expressed gene. Could the authors introduce another active gene (such as Gapdh) (full length, promoter only, gene body only - as done for alpha globin) to test whether any active gene is sufficient to block transcription?
- It is unclear how many replicates went into the Rad21 ChIP-seq.

Minor comments:

- The differences in Rad21 occupancy illustrated in Figure 4 are difficult to discern. Could the authors present the experimental data with the controls subtracted to make the differences clearer?
- Across the figures there is font inconsistency, and while some are legible (even too big) others are too small to read.

Referee #2:

In the presented manuscript, Cornell et al. investigate the nature of the 3' boundary of the alpha-globin locus as a model system for elucidating the relationship between chromatin structure and gene expression. They build on their previous elegant work on characterising the 5' boundary of this region. In this study, they utilised genetic mouse experiments both in vivo and in vitro and unexpectedly found that the CTCF-defined boundary elements play a minor role in the formation of the alpha-globin 3' sub-TAD structure and instead propose that Hba-a1 and Hba-a2 genes have this function, with Hba-a1 also serving as a boundary between the alpha-globin enhancers and Hba-a2. This is an exciting and important study, and its novelty lies in the demonstration that an active gene can act as a boundary element. The authors need to consider the following revisions to improve their study:

Major:

1. The authors suggest that active transcription and its machinery, rather than the DNA sequence itself, contribute to the function of the a1 gene as a boundary element. This needs to be tested by treating their cells with transcription initiation (triptolide) or transcription elongation (DRB) inhibitors.
2. To prove that the a1 gene is a CTCF-independent boundary element, they need to perform CTCF ChIP in their reporter cells to confirm the absence of CTCF binding in their inserts.
3. To demonstrate that a1 and a2 genes serve as boundary elements, the authors need to perform the inverse experiment and delete those genes, similarly to what they've done for CTCF binding sites or in their ES cell differentiation assay, to demonstrate

that the boundary is lost.

Minor:

4. Figure 3A - bring this diagram earlier to enhance understanding of the locus structure, as the figures before this one only have a very zoomed-out version.

5. The labelling of the many plots needs to be further improved to more precisely determine what is being plotted, i.e. 1D Fold change of what? , 1E percentage of what?

6. The Figure panels need to be re-labelled in the order they appear in the text. Figure 1E is referenced after all the panels in Figure 2. Figure 1D is mentioned before Figures 1B and 1C. Supplementary Figure 3 is referenced before Supplementary Figures 1 and 2.

7. Several long sentences need to be revised to improve the understanding/syntax, e.g., "Despite this coherent model integrating the role of enhancers, promoters, and insulators which relates genome structure to gene expression, not all CTCF-bound sites act as insulators (Dixon et al. 2012) and, importantly, not all TADs are flanked by convergent CTCF sites (Gomez-Marin et al. 2015; Rao et al. 2014)."

"Whilst CTCF is considered the primary insulator element in mammalian genomes, transcription has been previously associated with domain boundaries (Bonev et al. 2017; Dixon et al. 2012), reported to interact with the cohesin loop extrusion machinery (Busslinger et al. 2017) and with disruption of expression from downstream genes (Bozhilov et al. 2021; Cho et al. 2018)."

8. Figure 2C - add statistical analysis

9. Introduce sub-labels (e.g., A,B,C,D) to different panels/views of the Capture-C data and reference them in the text rather than having all the panels/views under one label.

10. In the following sentence, I assume the authors meant PolyA-unselected RNA. "To verify the transcriptional output from the inserted fragments, we performed PolyA selected RNA-seq from in vitro derived CD71+ cells from each of the models, allowing us to capture immature (PolyA-) transcripts from around the insertion site and mature transcripts (PolyA+)."

11. Is differentiation efficiency the same between different conditions in Figure 3? Do the authors measure YFP+ or YFP+CD71+?

Referee #3:

Cornell et al. investigate an important question in the 3D genome organization field on whether actively transcribed genes can serve as boundaries/insulators and delimit the action of enhancers. The authors build on their prior work on the alpha-globin locus and engineer careful perturbations, both in vitro and in vivo, to systematically dissect the effect of individual elements on chromatin contacts and transcription. They find that active transcription, rather than CTCF binding sites, behave as boundary elements in the context of alpha-globin gene regulation.

I really enjoyed reading this manuscript. I think that this study, and the systematically designed experiments, provides valuable insights to the field. It takes a lot of work to do the deletion experiments and I applaud the authors for doing the multiple deletions and combinations in vivo, coupled with Capture-C to measure the effects on structure but also collecting ChIP-seq, ATAC-seq and expression data to control for any confounding or incomplete effects and functionally measure the outcomes. This is really the strength of the paper. Where I struggled was the interpretation of the data and conclusions in some places. Below are specific comments that I think can be addressed with minimal additional experiments:

Main concerns:

1. If published and available, it would be nice to include a Hi-C map of the locus or a schematic of the Hi-C map. From the TAD/sub-TAD annotations and Capture C profiles, the main TAD boundary is delimited by HS -38/-39 CTCF peaks and sites near theta2/Hba-a2. I would also suggest expanding the view/or in a different figure to include Ubt2 as this gene was measured by qPCR.

2. While interesting, it is not obvious what effect is expected from deleting HS+44//48. As the authors highlighted in the introduction, only a minority of CTCF binding sites are TAD boundaries, and according to their TAD annotations and Capture-C profile, 44/48 aren't positioned at a TAD boundary. And so my expectation is that there wouldn't be an effect on the TAD comprising the alpha globin genes upon deletion of these sites. Perhaps the Hi-C or additional context could help understand why deletion of these sites in particular is relevant and if the result is surprising or expected. (Are they important in cells where the alpha globin genes are not actively transcribing?).

3. The results with theta sites and combined with 44/48 sites are very convincing in that these elements don't determine the 3' TAD boundary. I wonder, if in the erythroid cells, where the globin genes aren't expressed, if these elements do play a role in the

TAD boundary. Perhaps looking through published Hi-C data in other cell types to see if the TAD boundary is conserved could give some clues.

Choice of editor/authors: If not too tedious and the choice of cell type is clear, it would be interesting to see if the capture-c profile changes upon deletion of the theta + 44/48 sites, in a cell type that does not express the alpha-globin genes (which would suggest that these CTCF sites may still serve a role in other contexts).

4. It is not obvious for me from Sup. Fig 4A that Hba-a1 has a higher interaction frequency to enhancers than Hba-a2. If anything the different viewpoints in Sup. Fig 4A make it hard to compare directly. From Fig 1A and 1C, from the R2 viewpoint, it seems to me that the interaction frequencies to R2 are not that different between the two genes and are within measurement error. Given the very short distance between the two promoters (<4kb), I wouldn't expect these to be very different and if anything expected that the contact frequency decays slightly with distance.

The difference in transcription between the two genes could be a number of factors (maybe the slight differences in the promoter sequence), with increased stalling of cohesin at the first transcribed genes being one of the, the hypothesis the authors go on to elegantly test in the next Figure.

5. The boundary reporter experiment is very nice, and makes the point that introducing a strong insulator (HS38) or a highly transcribed gene, between the enhancer (R2) and the gene (Hba-a1), affects transcription. Although from Fig 4C, it doesn't look like the insertion of P+G altered the interaction frequency between R2 and Hba-a1.

Choice of editor/authors: I would be curious to see what the Capture-C profile looks like with the HS38 insertion relative to the P+G insertion. I would expect the insulator insertion to have an impact on contact frequency. But perhaps the experiment is too noisy here and the distances too small to see an effect.

6. Figure 3: a key control that is missing is to introduce the P+G insert downstream of Hba-venus to show that the changes are position dependent, and not due to inserting of a highly transcribed gene.

Can the authors also measure Hba-a2 expression, does it stay proportionally lower than a1?

Minor comments:

7. Introduction: The authors make the case that not all CTCF sites are found at TAD boundaries earlier in the paragraph and later describe how it is not fully understood if active transcription sites can serve as boundaries. However I find that the point about the role of CTCF and TAD boundaries in general for transcriptional regulation (lines 73-75, 82-83) is a separate question, and one that is outside of the scope of this paper and the point that the authors are making. I'd recommend removing these parts.

8. The figures aren't listed in order in the text. For example, there's a jump from 1A to 1D and 1E is referenced for the first time after the text describing Fig. 2.

9. In Fig 1D and 2D, does Hba stand for either Hba-a1 or a2? Or is Hbb supposed to represent Hba-a2 Or is Hbb = beta-globin and if so, can the authors explain why they measure beta-globin in this context?

10. Naive question: can the authors explain why an in vivo system (mouse model!) is required to assay the perturbations? Indeed the reporter assay in Figs. 3 and 4 uses an in vitro differentiation system and I am curious about the reasons behind the much more lengthy mouse model generation used to test the deletions of individual binding sites.

Referee #1:

In this manuscript, Cornell et al. investigate whether transcriptionally active genes can act as boundaries of 3D chromatin structures. Their study centres on the alpha globin gene which lies within a CTCF flanked TAD in mouse erythroid cells, where it is actively transcribed. They find deletion of the downstream CTCF motifs do not affect surrounding chromatin contacts or gene expression, suggesting the gene itself may define the boundary. In a previously validated insulation assay, they show the full-length alpha globin gene can act as an insulator, with the promoter or gene body alone showing modest insulator capabilities. This appears to be cohesin dependent, as it accumulates at the gene as we would expect it to at a CTCF boundary. Overall, the authors' careful dissection of this locus offers a new way to define boundaries independent of insulator proteins, opening the door to future studies into how widespread this phenomenon is.

I believe the paper would benefit from the following:

Major comments:

1. It is unclear whether the insulation potential of the alpha globin gene is unique or if the same would be seen for any highly expressed gene. Could the authors introduce another active gene (such as Gapdh) (full length, promoter only, gene body only - as done for alpha globin) to test whether any active gene is sufficient to block transcription?

In this manuscript, we have aimed to determine the 3' boundary of the functional domain of the alpha globin locus; following targeting of the CTCF sites, it appeared the α -globin genes are the primary delimiting element of this domain. We therefore wanted to specifically test the α -globin genes for their insulator/boundary activity.

Whilst we have not been able to test other promoters or genes in this case, we have examples of other genes acting in a similar manner (see lines 420-24). We have included a reference in which a PGK-Neo cassette inserted into the place of *Hba-x* (which is near the insertion site in these models) lead to a decrease in α -globin expression (Leder et al. 1997) (see lines 424-27). We have also previously described that a single nucleotide gain of function variant which forms a promoter between the enhancers and the native *HBA* genes, similarly, leads to a significant decrease in α -globin expression and development of alpha thalassemia. This is an instance of a non-native promoter causing changes in gene expression (Bozhilov et al. 2021).

In addition, some of the first reports describing characteristics of TAD boundaries noted the enrichment for the promoters of actively transcribed genes, such as housekeeping genes (Dixon et al. 2012; Hong and Kim 2017) – these observations are featured in the text at line 55-57. In addition, whilst under review, there have since been other studies reporting consistent findings to our study in the Sox2 locus. In the Sox2 locus, a similar conclusion that the Sox2 gene acts to restrict its enhancers' activity and appears to act as the 3' boundary of the functional domain (Eder et al. 2025). In a recent preprint, insertion of different promoters upstream of a reporter, with different inherent strengths found a correlation between the inserted promoter's strength and its ability to dampen the expression of a reporter gene

(Koska et al. 2025), similarly to our observation that the greater transcription of the promoter with gene inserted have greater insulation affect than the promoter only (Figure 3). These new studies have been referenced in the Discussion lines 400-405.

Therefore it is likely that other genes may have similar capabilities of having insulator-like activity and as the focus in this manuscript we have shown that the alpha globin genes are likely to be the primary domain 3' boundary. We have also discussed a potential mechanism, such that the genes may form partial barriers to cohesin mediated loop extrusion.

2. It is unclear how many replicates went into the Rad21 ChIP-seq.

Controls 2 samples, aP_{Only} 1 sample, aP+G 3 samples. This is now clarified in the figure legend as well as exemplified in the individual point data in Figure 4B.

Minor comments:

3. The differences in Rad21 occupancy illustrated in Figure 4 are difficult to discern. Could the authors present the experimental data with the controls subtracted to make the differences clearer?

Quantitation of reads in the insert region was performed on the Rad21 ChIP-seq, and this has been added as Figure 4B. In short, region counts were generated by using the deeptools MultibigwigSummary package from RPKM normalised bigwigs (binsize=1) from each of the Rad21 ChIP-seq datasets. Due to insufficient replicates for differential analysis by DESeq2, these counts were normalised to a region within the Beta-globin locus (mm9, chr7:110983901-111027888), which would be expected to have consistent read coverage between models and hence was used to normalise the counts across samples – this is added in Methods 620-24. From this the increase in cohesin over the insertion site is clear.

4. Across the figures there is font inconsistency, and while some are legible (even too big) others are too small to read.

We have now increased the size of text in the figures and standardised to Helvetica. In cases such as gene annotations and some axis annotations standardising to a larger font size was not possible but words are legible.

Referee #2:

In the presented manuscript, Cornell et al. investigate the nature of the 3' boundary of the alpha-globin locus as a model system for elucidating the relationship between chromatin structure and gene expression. They build on their previous elegant work

on characterising the 5' boundary of this region. In this study, they utilised genetic mouse experiments both in vivo and in vitro and unexpectedly found that the CTCF-defined boundary elements play a minor role in the formation of the alpha-globin 3' sub-TAD structure and instead propose that Hba-a1 and Hba-a2 genes have this function, with Hba-a1 also serving as a boundary between the alpha-globin enhancers and Hba-a2. This is an exciting and important study, and its novelty lies in the demonstration that an active gene can act as a boundary element. The authors need to consider the following revisions to improve their study:

Major:

5. The authors suggest that active transcription and its machinery, rather than the DNA sequence itself, contribute to the function of the a1 gene as a boundary element. This needs to be tested by treating their cells with transcription initiation (triptolide) or transcription elongation (DRB) inhibitors.

Whilst addition of these chemicals would indeed inhibit the transcription of the inserted genes, the addition of Triptolide and/or DRB would also affect the expression of the native (YFP tagged) gene, which is the primary read out of the insulation strength of the inserts. Therefore the use Pol II inhibitors would compromise the output of the insulator screen. In the place of this, the alpha-globin gene body (without its promoter) was used to check if there were sequence specific effects, uncoupled to transcription and indeed there was no effect on insulation strength. Therefore, we can conclude that the insulation effect seen is due to transcription and not underlying sequence.

6. To prove that the a1 gene is a CTCF-independent boundary element, they need to perform CTCF ChIP in their reporter cells to confirm the absence of CTCF binding in their inserts.

The α -globin sequences inserted are the exact native sequence of the α -globin genes, with two incorporated SNPs; there are no CTCF bound sites within the α -globin gene sequence or the promoter, as shown by the absence of peaks in CTCF ChIP-seq in WT Ter119+ cells in Figures 1-2 and so the inserts would equally not have any binding. CTCF ChIP-seq in Ter119+ plotted below, showing no binding over the regions that were inserted (see red regions in the figure below). Note the CTCF binding to the θ CTCF sites near to the α -globin genes.

To confirm that there were no CTCF binding sites in the inserts, a motif search was performed using fimo and the JASPAR position weight matrix for the CTCF consensus binding motif on the inserted sequences, including the small spacer sequences left over scars from cloning regions into the HDR vector and selection cassette removal. With a significance threshold of $p < 1e-4$, there were no CTCF motifs discovered in the inserted sequences. Therefore, we are confident that there is no significant CTCF binding to the inserted sequences.

Zoomed in CTCF ChIP-seq from primary erythrocyte material. The regions in red are equivalent to the sequences inserted into reporter cells. There is no significant binding of CTCF across the native sequences. The peaks to the right of the red regions are the $\theta 1/\theta 2$ CTCF sites.

7. To demonstrate that **a1 and a2 genes serve as boundary elements, the authors need to perform the inverse experiment and delete those genes**, similarly to what they've done for CTCF binding sites or in their ES cell differentiation assay, to demonstrate that the boundary is lost.

The functional domain we address in this manuscript is the small erythroid specific sub-TAD, which encapsulates and reflects the interaction between the alpha globin enhancers and promoters: deletion of the alpha globin genes would no doubt shrink this domain. We have previously investigated the domain structure in a model lacking both the α -globin genes (AMKO) in mouse foetal liver and found that a small domain encapsulating the enhancers could still be formed in the absence of the genes but no additional interactions were detected at the 3' boundary of the normal sub-TAD (Brown et al. 2018); however in this study we did not address in further detail the effect of the deletion of the genes on surrounding genes.

We do have unpublished data in a mouse model with a deletion spanning the *Hba-a1/2* genes, showing no perturbation of expression of the 3' genes (*Sh3pxd2b*, *Utd2*, *Efcad9*); suggesting that there is no gain of interactions with the genes 3' when the genes are removed. However, in the absence of the *Hba-a1/2*, the 5' genes are upregulated in this model, hence suggesting enhancer activity is redirected when their native target is absent – this highlights the complex nature of enhancer activity, and equally supports that the *Hba-a1/2* do normally draw enhancer activity to them and in an unexpected way help to delimit activity from the 5' of the locus.

Of interest, using the highly orthologous human alpha globin cluster we have previously reported that when both alpha globin genes are deleted, a gene (NME4)

lying over 100kb downstream of the cluster is upregulated, suggesting that, like the mouse alpha genes, the human alpha genes normally act as a 3'boundary to the alpha globin cluster (Lower et al. 2009).

Minor:

8. Figure 3A - bring this diagram earlier to enhance understanding of the locus structure, as the figures before this one only have a very zoomed-out version.

A simple schematic is now added to Fig 1. In addition, a new Figure EV1 has been added to show the domain and other ChIP-seq tracks of key factors in primary erythroid cells which will also be beneficial to the reader as an introduction to the locus' landscape.

9. The labelling of the many plots needs to be further improved to more precisely determine what is being plotted, i.e. 1D Fold change of what? ,1E percentage of what?

The axis and figure legends have been revised and reformatted. This is further clarified in the Methods at line 663-665.

10. The Figure panels need to be re-labelled in the order they appear in the text. Figure 1E is referenced after all the panels in Figure 2. Figure 1D is mentioned before Figures 1B and 1C. Supplementary Figure 3 is referenced before Supplementary Figures 1 and 2.

The notation of Figure 1 has been changed to reflect the order of appearance in the text. The content of the figure has also be adjusted to focus on the $\Delta\theta_1/\theta_2$ and $\Delta\text{HS}_{44/48}$ models with the $\Delta\theta_2$ moved to Fig EV3.

11. Several long sentences need to be revised to improve the understanding/syntax, e.g.,

The following changes have been made in the text in red:

(Line 45) "Despite this coherent model integrating the role of enhancers, promoters, and insulators relating genome structure to gene expression, there are exceptions. For example, not all CTCF-bound sites act as insulators (Dixon et al. 2012) and, importantly, not all TADs are flanked by convergent CTCF sites (Gomez-Marin et al. 2015; Rao et al. 2014)."

(Line 265) "Whilst CTCF is considered the primary insulator element in mammalian genomes, transcription has also been previously associated with domain boundaries (Bonev et al. 2017; Dixon et al. 2012); reported to interact with the cohesin loop extrusion machinery (Busslinger et al. 2017) ; and with disruption of expression from downstream genes (Bozhilov et al. 2021; Cho et al. 2018)."

12. Figure 2C - add statistical analysis

The results of a 2-way ANOVA have been included in the figure legend of 2C; there were no significant differences between blood counts between WT or $\Delta\theta 1/\theta 2/HS+44/HS+48$ mice.

13. Introduce sub-labels (e.g., A,B,C,D) to different panels/views of the Capture-C data and reference them in the text rather than having all the panels/views under one label.

For panels with more than one Capture-C track, sub-labels have been added to improve the ease of reading similar panels.

14. In the following sentence, I assume the authors meant PolyA-unselected RNA. "To verify the transcriptional output from the inserted fragments, we performed PolyA selected RNA-seq from in vitro derived CD71+ cells from each of the models, allowing us to capture immature (PolyA-) transcripts from around the insertion site and mature transcripts (PolyA+)."

We prepared both PolyA Plus and PolyA Minus RNA for sequencing, this sentence has been adjusted in text to clarify this (Line 296).

15. Is differentiation efficiency the same between different conditions in Figure 3? Do the authors measure YFP+ or YFP+CD71+?

We did take measures from CD71+YFP+ cells, this has been clarified in Figure 3 by providing an example of the gating strategy used. The mESC-EB differentiation yields erythroid CD71+ cells in frequencies ranging between (15-50% of gated single cells). This range of variation is corrected for by measuring YFP fluorescence in only CD71+ cells, as described in the text (line 279, and Methods line 560-2).

Referee #3:

Cornell et al. investigate an important question in the 3D genome organization field on whether actively transcribed genes can serve as boundaries/insulators and delimit the action of enhancers. The authors build on their prior work on the alpha-globin locus and engineer careful perturbations, both in vitro and in vivo, to systematically dissect the effect of individual elements on chromatin contacts and transcription. They find that active transcription, rather than CTCF binding sites, behave as boundary elements in the context of alpha-globin gene regulation.

I really enjoyed reading this manuscript. I think that this study, and the systematically designed experiments, provides valuable insights to the field. It takes a lot of work to do the deletion experiments, and I applaud the authors for doing the multiple deletions and combinations in vivo, coupled with Capture-C to measure the effects on structure but also collecting ChIP-seq, ATAC-seq and expression data to control for any confounding or incomplete effects and functionally measure the outcomes. This is really the strength of the paper. Where I struggled was the interpretation of the data and conclusions in some places. Below are specific comments that I think can be addressed with minimal additional experiments:

Main concerns:

16. If published and available, it would be nice to include a Hi-C map of the locus or a schematic of the Hi-C map. From the TAD/sub-TAD annotations and Capture C profiles, the main TAD boundary is delimited by HS -38/-39 CTCF peaks and sites near theta2/Hba-a2. I would also suggest expanding the view/or in a different figure to include *Ubt2* as this gene was measured by qPCR.

A new figure EV1, has been added to display the locus as suggested by the reviewer. This figure shows a Tiled-C map of interactions at a 2kb resolution in WT *ter119+* erythroid cells adapted from (Oudelaar et al. 2020), with additional tracks from the same cells to orientate the reader. This figure has an expanded view of the locus which shows the position of 3' gene *Ubt2*. We have opted not to expand the figures within the main text as there are no informative changes of the domain past this region, and the Capture-C data will be more compressed to do so and it may impact interpretation.

17. While interesting, it is not obvious what effect is expected from deleting HS+44/48. As the authors highlighted in the introduction, only a minority of CTCF binding sites are TAD boundaries, and according to their TAD annotations and Capture-C profile, 44/48 aren't positioned at a TAD boundary. And so my expectation is that there wouldn't be an effect on the TAD comprising the alpha globin genes upon deletion of these sites. Perhaps the Hi-C or additional context could help understand why deletion of these sites in particular is relevant and if the result is surprising or expected. (Are they important in cells where the alpha globin genes are not actively transcribing?).

We have previously performed tiled capture-C (Tiled-C) to generate pairwise interaction heatmaps to a ~2 kb resolution at various stages through erythroid differentiation, spanning mESCs, early erythroid progenitors (S0) to terminally differentiated mature erythroid cells (S4) (Oudelaar et al. 2020). In mESCs and throughout all stages of differentiation, the larger ~165 kb TAD can be observed, this larger TAD does appear to be CTCF-CTCF delimited, and by the HS+44/48 CTCF sites (See figure below). We have included the regions corresponding to the larger TAD and erythroid specific sub-TAD as bars above all Capture-C tracks presented. For clarity, we have included a new Supplementary Figure 1 to clarify where these structures can be visualised. It is worth noting that this ~165 kb TAD is located far from other neighbouring domains in *cis* and therefore we wouldn't expect the HS+44/48 deletions to affect other neighbouring loci (Oudelaar et al. 2020).

Supplementary Figure 2: Tiled-C contact matrices of the α -globin locus in erythroid and ES cells.

Figure from (Oudelaar et al. 2020). The top panel shows tiled-C data across the alpha globin locus from erythroid cells, as a pairwise heatmap of interactions at a ~2kb resolution. The elements of the locus are highlighted in ATAC and CTCF ChIP-seq tracks underneath. The lower panel shows the same region but in mESCs – not that the enhancers and genes

are not active in mESCs and only the larger TAD domain, with specific CTCF-CTCF interactions is observed.

In addition, specific CTCF-CTCF interactions are highlighted in both mESCs and erythroid cells as addressed by our high resolution MCC (Hua et al. 2021). These CTCF-CTCF interactions form the larger cell type invariant TAD and therefore they are likely functional in keeping the extremities of the chromosomal region together by genome-wide cohesin loading, similarly to that proposed in (Karpinska et al. 2025). Nested within this larger TAD, there is a smaller ~65 kb erythroid-specific sub-TAD, which represents the interactions between the enhancers and promoters, and are those observed in Capture-C tracks in this manuscript and delimited by the HS-38/39 CTCF. We consider this the functional domain and aim to assess which elements are significant in forming this structure in this manuscript.

Figure from (Hua et al. 2021). High resolution Micro Capture-C (MCC – purple tracks) from the viewpoints of the CTCF sites in erythroid cells (top panel) and mESCs (lower panel). Note the specific CTCF-CTCF interactions between the HS-39 CTCF site and the 3’CTCF sites (pink and purple arrows) which are present but to a lesser level in mESC compared to erythroid cells. Dnase hypersensitivity, CTCF and Rad21 tracks display the positions of elements.

As $\theta 1/\theta 2$ CTCF site deletions didn’t influence α -globin expression we of course wanted to rule out that HS+44/48 were functional in regulating the α -globin genes and due to their position in the larger TAD, we wanted to determine their role in forming this boundary. With the HS+44/48 deletion there is no effect on α -globin expression and therefore we can consider the CTCF-CTCF interactions to be acting as a support for the enhancer-promoter by perhaps maintaining a larger, weaker TAD domain, but this support is not essential for proper establishment of expression. Equally, as this was performed in a mouse, the enhancer-promoter contacts have

developed correctly over development in the absence of the HS+44/48 and so further highlighting that they are not essential to locus function.

18. The results with theta sites and combined with 44/48 sites are very convincing in that these elements don't determine the 3' TAD boundary. I wonder, if in the erythroid cells, where the globin genes aren't expressed, if these elements do play a role in the TAD boundary. Perhaps looking through published Hi-C data in other cell types to see if the TAD boundary is conserved could give some clues.

As discussed above, the outer HS+44/48 do appear at the boundaries of the larger, non-erythroid TAD and they do form specific contacts with other CTCF bound sites in the locus in both erythroid and mESCs as assessed by MCC (Hua et al. 2021) and by Tiled-C (Oudelaar et al. 2020). However, the erythroid specific interactions of the subTAD (HS-38/39, enhancers and genes) do not seem to rely on the HS+44/48 sites.

19. Choice of editor/authors: If not too tedious and the choice of cell type is clear, it would be interesting to see if the capture-c profile changes upon deletion of the theta +44/48 sites, in a cell type that does not express the alpha-globin genes (which would suggest that these CTCF sites may still serve a role in other contexts).

With the discussion above, the $\theta 1/\theta 2/+44/48$ sites do form CTCF-CTCF interactions in mESC and throughout erythroid differentiation, from early to late stages, but as shown in this manuscript, these interactions are not essential for establishment of correct globin enhancer-promoter interaction.

The CTCF-CTCF interactions may form due to cohesin translocation and stalling specifically at these sites and aiding the formation of chromatin hubs as discussed in (Karpinska et al. 2025). Therefore, the interactions are present in other cell types, however, do not appear to be functional in gene regulation.

20. It is not obvious for me from Sup. Fig 4A that Hba-a1 has a higher interaction frequency to enhancers than Hba-a2. If anything, the different viewpoints in Sup. Fig 4A make it hard to compare directly. From Fig 1A and 1C, from the R2 viewpoint, it seems to me that the interaction frequencies to R2 are not that different between the two genes and are within measurement error. Given the very short distance between the two promoters (<4kb), I wouldn't expect these to be very different and if anything expected that the contact frequency decays slightly with distance.

Hba-a1/2 specific Capture-C is made possible by the presence of a SNP upstream of the promoter located near to a DpnII recognition site. It is possible to enrich for interactions *Hba-a1/2* by using a capture oligo specific to the DpnII fragment over the promoters and bioinformatically extracting reads mapping to this fragment containing the base specific to either variant. They can only be distinguished when using a probe to enrich for the promoter (i.e. from the Hba viewpoint) because, due to the SNP-specific filtering, there is a considerable loss of usable reporter reads and so there is a reduction in the depth of the data, and hence data appears noisier. The

reviewer was correct in identifying the noise in this experiment and we have revisited the data from the SNP-specific capture traces in Supplementary Figure 4A ($\Delta\theta_2$, $\Delta\theta_1/\theta_2$) and determined that, due to this filtering, the read depth was indeed not of a sufficient level to draw interpretation from (totalling only ~2000-6000 reporters in replicates) and so we have removed these plots. However, we have also determined that the data from the $\Delta\theta_1/\theta_2/44/48$ had a more sufficient depth (~12000-15000 reporters) to perform this analysis, and this has been included in the place of the $\Delta\theta_2$, $\Delta\theta_1/\theta_2$ data. In these variant specific profiles, *Hba-a1* interacts more broadly with the enhancers compared to *Hba-a2*; in the absence of the 4 CTCF sites, the profiles are identical when comparing the WT, hence there is no effect on the preferential interactions between the genes.

Regarding all other figures (Figure 1,2,4) with Capture-C profiles from all other viewpoints (enhancers and CTCF sites), the Capture-C reporter reads over the *Hba-a1/2* genes are not variant specific and allowed us to map equally to both copies in the reference genome; the resulting signal plotted is an amalgamation of reporters from the two genes. We therefore do not interpret any difference in interactions between the *Hba-a1/2* genes when mapping interactions from enhancer or CTCF site viewpoints and only interpret differential interactions when using the SNP variant specific approach above.

21. The difference in transcription between the two genes could be a number of factors (maybe the slight differences in the promoter sequence), with increased stalling of cohesin at the first transcribed genes being one of the hypothesis, the authors go on to elegantly test in the next Figure.

No corrections required.

22. The boundary reporter experiment is very nice, and makes the point that introducing a strong insulator (HS38) or a highly transcribed gene, between the enhancer (R2) and the gene (*Hba-a1*), affects transcription. Although from Fig 4C, it doesn't look like the insertion of P+G altered the interaction frequency between R2 and *Hba-a1*.

The Capture-C was repeated in revision to produce more complex 3C libraries presented in Figure 4C with quantification of regions flanking sites of interest specifically quantified from the locus wide profile. Whilst there are subtle differences in these regions, interaction profiles from this model are complicated to interpret due to the similarity between the inserted sequences and that of the native genes, which means the DpnII fragment containing the most critical *Hba*-promoter interaction is impossible to assign to either the insert or the native gene; this has been described in an added section of supplementary material. With this limitation in mind, the evidence that there is a gain of accessibility at the site of insertion, with increased transcription and a reduction in native gene expression all indicates that the inserted sequences are impacting normal enhancer-promoter communication.

23. Choice of editor/authors: I would be curious to see what the Capture-C profile looks like with the HS38 insertion relative to the P+G insertion. I would expect the insulator insertion to have an impact on contact frequency. But perhaps the experiment is too noisy here and the distances too small to see an effect.

We have a paper in revision that answers this question directly – we insert the HS-38 CTCF site in a similar position to the insertion site here in different orientations in mice and in primary erythroid cells. We find that there is a significant reduction in interactions between the enhancers and the *Hba-a1/2* genes when the HS-38 site introduced into a position between the enhancers and the native genes (Stolper et al. 2023). As discussed in the new limitations section of the Supplementary, due to the sequence similarity between the inserted sequences and the targets, it is impossible to distinguish reporter reads from the insert or the native sequences, and so this may be masking differences in interactions at the *Hba-a1/2* genes.

24. Figure 3: a key control that is missing is to introduce the P+G insert downstream of *Hba-venus* to show that the changes are position dependent, and not due to inserting of a highly transcribed gene.

We would like to highlight that the position of *Hba-a2* relative to the *Hba-a1* gene is the equivalent of this proposed experiment. As discussed in the text and in Figure 1, *Hba-a2* consistently produces ~30% of total α -globin transcripts compared to *Hba-a1* producing ~70%. Multiplications of the gene in other models led to a gradient of expression with those proximal to the enhancers been more transcribed than those found sequentially more distal (Liebhaber, Cash, and Main 1985; Liebhaber and Kan 1981; Orkin and Goff 1981; Vestri, Pieragostini, and Ristaldi 1994) – referenced in the text lines 447-450. We therefore can predict with some certainty that additional insertions of the α -globin gene downstream would not likely be lesser transcribed, either due to the presence of the other genes or due to distance decay affecting communication with enhancers. However, in the human locus, the 2 identical copies of the gene are found only 4kb apart, and so distance decay should not affect the expression, yet the 70/30 ratio of transcripts are still observed. All this evidence suggests a clear positional effect on α -globin expression. In our reporter models, *Hba-a2* is present downstream of the *Hba-a1:Venus*; as will be discussed in the following comment, we unfortunately could not read-out *Hba-a2* in the reporter models.

The experiment proposed would be in attempt to address the question of promoter competition and, as we discuss in the text, this cannot be ruled out (see lines 414-20). The inserted copy is expressed to a far greater level than the native genes, likely due to the increased proximity to the enhancer cluster, and it is feasible that a massively transcribing gene would sequester transcriptional machinery from other targets. We suggest there is an interaction with loop extrusion as there are other lines of evidence suggesting that cohesin is important in the context of this locus and insertion site, as we observe a clear significant dependence of insulator strength on the orientation of CTCF sites (Stolper et al. 2023; Tsang et al. 2024).

25. Can the authors also measure *Hba-a2* expression, does it stay proportionally lower than *a1*?

SNP-specific counting was performed in the RNA-seq from the insertion/reporter models and found that all RNA-seq reads mapped to the *Hba-a1* base in all the insertion lines derived from the WT- α YFP hemizygous parental line. To determine why this was observed, a long-range PCR (specific to the *Hba-a2* region) was

performed and sequenced using Nanopore sequencing – the resulting sequence alignment to mm10 is included in the Figure below. We discovered that 2 SNPs that were expected to be specific to *Hba-a1::Venus* (one the natural SNP in exon 3 and another 8bp upstream of the natural SNP introduced as a PAM site inactivating mutation following introduction of *mVenus*) appear to have been introduced into the *Hba-a2* sequence. *Hba-a2* itself was not targeted with *mVenus* but as the *Hba-a1/2* sequences are identical (excluding the expected SNP in exon 3) it is possible that during introduction of the *mVenus* at exon 3 of *Hba-a1*, the guide directing CRISPR-Cas9 recognised exon 3 of the *Hba-a2*, created a break that was then repaired leading the SNPs in exon 3 to be corrected with a small homologous region of *Hba-a1*. Therefore, whilst the sequence of *Hba-a2* is intact and it is in no doubt transcribed; we cannot distinguish between *Hba-a1/2* bioinformatically due to the loss of the SNP in *Hba-a1::Venus* reporter cells and their derivatives. We further cannot take action to correct this to determine read out for fear of disrupting *Hba-a1* and the inserted sequences. Finally, we cannot assess changes by rt-qPCR as primers across *Hba-a2* and the inserted sequences would cross react.

Whilst we cannot observe the gradient of transcription from the insertion models we would like to highlight the other evidence in the text which corroborates the position dependant expression of multiplied copies of α -globin in other models as discussed above. In addition, we were able to count reads specifically mapping to the inserted copy of the α -globin gene (α P+G) due to the inclusion of a different SNP in exon 3; as expected, the inserted α -globin composes most transcripts compared to the native genes. This is now featured in Fig EV4.

Figure: Alignment of *Hba-a2* specific PCR using long read sequencing reveals loss of identifying SNP in reporter lines. Reads from a long-range PCR (~5kb) specific to the *Hba-a2* aligned to mm10; exon 3 of the gene where the SNP is expected is highlighted in the black box. The product from E14 WT gDNA shows the base composition is of the expected *Hba-a2* reference. Reads from representative clones from each reporter model line are shown below, showing all reads contain *Hba-a1* SNPs (green and blue bars). These are not from *Hba-a1* as the PCR is specific to *Hba-a2* and reads lack the *Venus* sequence.

Exonic SNP-specific count PolyA-Plus RNA-seq. The αP+G gene had its own unique SNP in exon 3 allowing counting of the proportion of transcripts between the inserted or native α-globin copies. SNPs specific counts of each variant of *Hba* were counted similarly to Figure 1E in RNA-seq from the CD71+ cells.

Minor comments:

26. Introduction: The authors make the case that not all CTCF sites are found at TAD boundaries earlier in the paragraph and later describe how it is not fully understood if active transcription sites can serve as boundaries. However, I find that the point about the role of CTCF and TAD boundaries in general for transcriptional regulation (lines 73-75, 82-83) is a separate question, and one that is outside of the scope of this paper and the point that the authors are making. I'd recommend removing these parts.

We would like to argue that these points are relevant to the manuscript; here we assess the functional role of specific CTCF bound sites in delimiting enhancer-promoter interactions and hence their role in transcriptional regulation. The observations we reference provide the context that whilst CTCF does instruct 3D structure and TAD domains, it is still unclear what role CTCF sites play in gene regulation. We reference conflicting findings, such that global depletion of CTCF does not lead to significant changes in gene expression, but in some specific cases when CTCF sites at TAD boundaries are deleted they do appear to be functional as insulators.

27. The figures aren't listed in order in the text. For example, there's a jump from 1A to 1D and 1E is referenced for the first time after the text describing Fig. 2.

This has been corrected in the Figure and Text

28. In Fig 1D and 2D, does Hba stand for either Hba-a1 or a2? Or is Hbb supposed to represent Hba-a2 Or is Hbb = beta-globin and if so, can the authors explain why they measure beta-globin in this context?

Hba is a combination of both *Hba-a1* and *Hba-a2*, due to their sequence similarity it is difficult to distinguish between them using rt-qPCR and so they are considered as one signal. Hbb represents the summation of adult beta-globin (*Hbb-b1*, *Hbb-b2*) similarly due to their sequence similarity. As alpha globin and beta globin are expected to be found at similar ratios in normal erythroid cells, we plot the *Hba:Hbb*

ratio to determine if there is a shift in Hba expression. A sentence has been added at the end of the Methods RNA expression analysis to explain how the ratio was calculated (Line 665-666).

29. Naive question: can the authors explain why an *in vivo* system (mouse model!) is required to assay the perturbations? Indeed the reporter assay in Figs. 3 and 4 uses an *in vitro* differentiation system and I am curious about the reasons behind the much more lengthy mouse model generation used to test the deletions of individual binding sites.

We first generated the mouse models as at the time as previous complementary experiments assessing the 5' CTCF sites (Hanssen et al. 2017) were done in primary erythroid cells and so was to be able to compare the outcome of these CTCF site perturbations in the same system. It is also beneficial to assess how these genetic perturbations affect gene regulation throughout development which is possible in mouse models. Whilst these data are very valuable, as the reviewer rightly comments, generating mouse models is laborious, lengthy and costly – hence the need for the *in vitro* differentiation system.

As the *in vitro* differentiation system was verified (Francis et al. 2022), we were then able to expand the number of genetic models we could generate, without losing the ability to test perturbations in erythroid cells, therefore allowing a larger scaling of insertions, as shown in (Tsang et al. 2024). Hence, the switch to using the *in vitro* system for the reporter assay.

- Bozhilov, Yavor K., Damien J. Downes, Jelena Telenius, A. Marieke Oudelaar, Emmanuel N. Olivier, Joanne C. Mountford, Jim R. Hughes, Richard J. Gibbons, and Douglas R. Higgs. 2021. 'A gain-of-function single nucleotide variant creates a new promoter which acts as an orientation-dependent enhancer-blocker', *Nature Communications*, 12.
- Brown, J. M., N. A. Roberts, B. Graham, D. Waithe, C. Lagerholm, J. M. Telenius, S. De Ornellas, A. M. Oudelaar, C. Scott, I. Szczerbal, C. Babbs, M. T. Kassouf, J. R. Hughes, D. R. Higgs, and V. J. Buckle. 2018. 'A tissue-specific self-interacting chromatin domain forms independently of enhancer-promoter interactions', *Nat Commun*, 9: 3849.
- Dixon, J. R., S. Selvaraj, F. Yue, A. Kim, Y. Li, Y. Shen, M. Hu, J. S. Liu, and B. Ren. 2012. 'Topological domains in mammalian genomes identified by analysis of chromatin interactions', *Nature*, 485: 376-80.
- Eder, M., C. J. I. Moene, L. Dauban, M. Magnitov, J. Drayton, M. de Haas, C. Leemans, M. Verkuilen, E. de Wit, A. S. Hansen, and B. van Steensel. 2025. 'Functional maps of a genomic locus reveal confinement of an enhancer by its target gene', *Science*, 389: eads6552.
- Francis, Helena S., Caroline L. Harold, Robert A. Beagrie, Andrew J. King, Matthew E. Gosden, Joseph W. Blayney, Danuta M. Jeziorska, Christian Babbs, Douglas R. Higgs, and Mira T. Kassouf. 2022. 'Scalable *in vitro* production of defined mouse erythroblasts', *Plos One*, 17.
- Hanssen, L. L. P., M. T. Kassouf, A. M. Oudelaar, D. Biggs, C. Preece, D. J. Downes, M. Gosden, J. A. Sharpe, J. A. Sloane-Stanley, J. R. Hughes, B. Davies, and D. R. Higgs. 2017. 'Tissue-specific CTCF-cohesin-mediated chromatin architecture delimits enhancer interactions and function *in vivo*', *Nat Cell Biol*, 19: 952-61.

- Hong, S., and D. Kim. 2017. 'Computational characterization of chromatin domain boundary-associated genomic elements', *Nucleic Acids Res*, 45: 10403-14.
- Hua, Peng, Mohsin Badat, Lars L. P. Hanssen, Lance D. Hentges, Nicholas Crump, Damien J. Downes, Danuta M. Jeziorska, A. Marieke Oudelaar, Ron Schwessinger, Stephen Taylor, Thomas A. Milne, Jim R. Hughes, Doug R. Higgs, and James O. J. Davies. 2021. 'Defining genome architecture at base-pair resolution', *Nature*, 595: 125-29.
- Karpinska, M. A., Y. Zhu, Z. Fakhraei Ghazvini, S. Ramasamy, M. Barbieri, T. B. N. Cao, N. Varahram, A. Aljahani, M. Lidschreiber, A. Papantonis, and A. M. Oudelaar. 2025. 'CTCF depletion decouples enhancer-mediated gene activation from chromatin hub formation', *Nat Struct Mol Biol*, 32: 1268-81.
- Koska, M., T. Swigut, A. N. Boettiger, and J. Wysocka. 2025. 'Promoter strength and position govern promoter competition', *bioRxiv*.
- Leder, Aya, Cathie Daugherty, Barry Whitney, and Philip Leder. 1997. 'Mouse ζ - and α -Globin Genes: Embryonic Survival, α -Thalassemia, and Genetic Background Effects', *Blood*, 90: 1275-82.
- Liebhaber, S. A., F. E. Cash, and D. M. Main. 1985. 'Compensatory increase in alpha 1-globin gene expression in individuals heterozygous for the alpha-thalassemia-2 deletion', *J Clin Invest*, 76: 1057-64.
- Liebhaber, S. A., and Y. W. Kan. 1981. 'Differentiation of the mRNA transcripts originating from the alpha 1- and alpha 2-globin loci in normals and alpha-thalassemics', *J Clin Invest*, 68: 439-46.
- Lower, K. M., J. R. Hughes, M. De Gobbi, S. Henderson, V. Viprakasit, C. Fisher, A. Goriely, H. Ayyub, J. Sloane-Stanley, D. Vernimmen, C. Langford, D. Garrick, R. J. Gibbons, and D. R. Higgs. 2009. 'Adventitious changes in long-range gene expression caused by polymorphic structural variation and promoter competition', *Proc Natl Acad Sci U S A*, 106: 21771-6.
- Orkin, S. H., and S. C. Goff. 1981. 'The duplicated human alpha-globin genes: their relative expression as measured by RNA analysis', *Cell*, 24: 345-51.
- Oudelaar, A. Marieke, Robert A. Beagrie, Matthew Gosden, Sara de Ornellas, Emily Georgiades, Jon Kerry, Daniel Hidalgo, Joana Carrelha, Arun Shivalingam, Afaf H. El-Sagheer, Jelena M. Telenius, Tom Brown, Veronica J. Buckle, Merav Socolovsky, Douglas R. Higgs, and Jim R. Hughes. 2020. 'Dynamics of the 4D genome during in vivo lineage specification and differentiation', *Nature Communications*, 11: 2722.
- Stolper, Rosa J., Felice H. Tsang, Emily Georgiades, Lars L. P. Hansen, Damien J. Downes, Caroline L. Harrold, Jim R. Hughes, Robert A. Beagrie, Benjamin Davies, Mira T. Kassouf, and Douglas R. Higgs. 2023. 'Loop extrusion by cohesin plays a key role in enhancer-activated gene expression during differentiation', *bioRxiv*.
- Tsang, F. H., R. J. Stolper, M. Hanifi, L. J. Cornell, H. S. Francis, B. Davies, D. R. Higgs, and M. T. Kassouf. 2024. 'The characteristics of CTCF binding sequences contribute to enhancer blocking activity', *Nucleic Acids Res*, 52: 10180-93.
- Vestri, R., E. Pieragostini, and M. S. Ristaldi. 1994. 'Expression gradient in sheep alpha alpha and alpha alpha alpha globin gene haplotypes: mRNA levels', *Blood*, 83: 2317-22.

Dear Dr. Kassouf,

Thank you for submitting a revised version of your manuscript. Your study has now been seen by all original referees, who find that most of their previous concerns have been addressed and now recommend publication of the manuscript after some additional minor revisions. I find these remaining minor concerns constructive and reasonable and would therefore ask you to incorporate the requested changes into the final version of the manuscript. In particular, I concur with referee #2 that the responses to concerns #6 and #7 should also be reflected in the manuscript. However, according to our policy, a reference to "data not shown" is not allowed and it might be therefore better to only refer to the published studies in paragraph 1 and 3 of your response to point #7.

In addition, there remain only a few mainly editorial points that have to be addressed before I can extend formal acceptance of the manuscript:

- MANUSCRIPT FORMAT:(no figures!) .docx, figures should be removed from ms, they should only be upldd as individual Figure files, no track changes, comments in
- FUNDING INFO: funding info included in the Comments box could not be extracted by Production team, and therefore all funders should be added to "More Funders" list
- On the abstract page of the manuscript, please include 4-5 general keyword terms to enhance searchability.
- As we are switching from a free-text author contribution statement towards a more formal statement based on Contributor Role Taxonomy (CRediT) terms, please remove the present Author Contribution section and instead specify each author's contribution(s) directly in the Author Information page of our submission system during upload of the final manuscript. See <https://casrai.org/credit/> for more information.
- FIGURE CALLOUTS: all callouts should be listed sequentially; there is a callout for Fig. 1F, but no such panel in Fig. 1; missing callout for Fig. 1D
- Figure nomenclature should be Figure EV1-EV4 instead of Extended Figure 1-4
- APPENDIX 1 FILE WITH ToC: nomenclature should be Appendix Figure Sx and Appendix Table Sx throughout ms and Appendix PDF
- R&T TABLE: in, but should be removed from ms, it's enough to be upldd individually
- SYNOPSIS TEXT: in, but Synopsis text seems too long. The 'blurb' text prefacing and summing up the conceptual aspect of the study should be no more than two sentences (max. 250 characters), followed by 3-5 one-sentence 'bullet points' with brief factual statements of key results of the paper
- Please note that the specific URLs for GSE153209, GSE97871, GSE137477 datasets are not provided in the data availability statement.
- Figure Legends (main + EV):
 1. Please note that the exact p values are not provided in the legends of figures 3B; EF 4 A-C
 2. Please note that the error bars are not defined in the legends of figures EV4 D, E
- Author email bounced:
 - Samy Alghadban - samy.alghadban@bms.ox.ac.uk
 - Damien Downes - damien.downes@ndcls.ox.ac.uk
 - Rosa Stolper - rosa.stolper@well.ox.ac.uk
- Sections need to be named and the order should be corrected: Title page - Abstract - Keywords - Introduction - Results - Discussion - Methods - Data Availability - Acknowledgements - Disclosure and Competing Interests Statement - References - Figure Legends - Table(s) - Expanded View Figure Legends.

With best regards,

Cornelius Schneider

Cornelius Schneider, PhD
Editor | The EMBO Journal
c.schneider@embojournal.org

Please refer to our figure preparation guideline in order to ensure proper formatting and readability in print as well as on screen:

See also figure legend guidelines:

<https://www.embopress.org/page/journal/14602075/authorguide#figureformat>

Referee #1:

The revised paper addressed all the raised concerns.

Referee #2:

The authors sufficiently addressed the comments I raised in their point-by-point response. However, I would prefer to see the reply to two of my major comments incorporated in the manuscript as well, not just in the point-by-point response. In particular, this relates to comments #6 and #7. In response to comment 7, the authors refer to the unpublished data in the mouse model with the deletion of the Hba-a1/2 genes. Why wasn't this included in the revised manuscript? They performed motif analysis in response to comment 6, why wasn't this included in the manuscript? If these points are addressed, I am happy with the manuscript being accepted for publication.

Referee #3:

I thank the authors for addressing the concerns and revising the manuscript. This work, with the systematic deletions of CTCF sites and the reporter assay elegantly show that active transcription can serve as an insulator. This is a very valuable insight for the field. The revised text and figures as well as the new analyses and experiments address all my concerns.

I have a few minor, mainly esthetic, suggestions but otherwise recommend this manuscript for publication.

1) Please check the figure order and references in text:

- Line 159: what does Fig. S1 refer to?
- Fig 1E is referenced after Fig 2
- Line 172: There is no figure 1F (did the authors mean 1D ?)
- EV2B comes after EV3 in text
- Figure 3F is not cited

2) Line 228-230. I think the authors mean to reference Fig 2EVB iii and iv and not Fig 3EVB iii, iv.

I can now see that comparing the values on the y axis between iii and iv shows that Hba-a1 has a higher interaction frequency with R1 going from ~40% from Hba-a1 (iii) to ~20% in than Hba-a2 (iv). It is not obvious that the values are different for the other enhancers. Can the authors plot the quantification of the values in a bar plot (or another way to plot) for each enhancer to make it easier to compare side by side?

3) It is helpful to have Fig EV1 with the TAD structure and all the genes in the locus. If the resolution allows, can the authors also add a zoom-in version for the TAD (region under the black bar in the current EV1) as well? This may also help with the Tiled-C heatmap scaling as now, all the areas outside of the TAD are very dim.

Dear Prof Schneider,

We have now addressed all your and the reviewers concerns as detailed below.

Best Wishes,
Mira

#####

Thank you for submitting a revised version of your manuscript. Your study has now been seen by all original referees, who find that most of their previous concerns have been addressed and now recommend publication of the manuscript after some additional minor revisions. I find these remaining minor concerns constructive and reasonable and would therefore ask you to incorporate the requested changes into the final version of the manuscript. In particular, I concur with referee #2 that the responses to concerns #6 and #7 should also be reflected in the manuscript. However, according to our policy, a reference to "data not shown" is not allowed and it might be therefore better to only refer to the published studies in paragraph 1 and 3 of your response to point #7. In addition, there remain only a few mainly editorial points that have to be addressed before I can extend formal acceptance of the manuscript:

- MANUSCRIPT FORMAT:(no figures!) .docx, figures should be removed from ms, they should only be uploaded as individual Figure files, no track changes, comments in

Figures have been removed from the .docx

- FUNDING INFO: funding info included in the Comments box could not be extracted by Production team, and therefore all funders should be added to "More Funders" list

Acknowledgements have been removed from the More Funders box on the portal – they are solely referenced in the Acknowledgements now.

- On the abstract page of the manuscript, please include 4-5 general keyword terms to enhance searchability.

Keywords have been added under the abstract: Insulators, Transcription, Regulation, CTCF, Domains

- As we are switching from a free-text author contribution statement towards a more formal statement based on Contributor Role Taxonomy (CRediT) terms, please remove the present Author Contribution section and instead specify each author's contribution(s) directly in the Author Information page of our submission system during upload of the final manuscript. See <https://casrai.org/credit/> for more information.

On portal

- FIGURE CALLOUTS:

all callouts should be listed sequentially;

there is a callout for Fig. 1F, but no such panel in Fig. 1; missing callout for Fig. 1D

This has been corrected, 1F was incorrectly called out in the place of 1D. 1E has also been referred to earlier in the text.

- Figure nomenclature should be Figure EV1-EV4 instead of Extended Figure 1-4

This has been corrected; all EV nomenclature has been checked and changed where needed.

- APPENDIX 1 FILE WITH ToC: nomenclature should be Appendix Figure Sx and Appendix Table Sx throughout ms and Appendix PDF

This has been corrected: all Appendix nomenclature has been checked and changed where needed.

- R&T TABLE: in, but should be removed from ms, it's enough to be uploaded individually

The reagents table has been removed from the .docx

- SYNOPSIS TEXT: in, but Synopsis text seems too long. The 'blurb' text prefacing and summing up the conceptual aspect of the study should be no more than two sentences (max. 250 characters), followed by 3-5 one-sentence 'bullet points' with brief factual statements of key results of the paper

The synopsis text has been reduced 250 characters.

- Please note that the specific URLs for GSE153209, GSE97871, GSE137477 datasets are not provided in the data availability statement.

Hyperlinks have been added to link to each of the accession numbers in the .docx

If needed, the full links can be found below:

<https://www.ncbi.nlm.nih.gov/geo/query/acc.cgi?acc=GSE153209>

<https://www.ncbi.nlm.nih.gov/geo/query/acc.cgi?acc=GSE313786>

<https://www.ncbi.nlm.nih.gov/geo/query/acc.cgi?acc=GSE97871>

<https://www.ncbi.nlm.nih.gov/geo/query/acc.cgi?acc=GSE137477>

<https://www.ncbi.nlm.nih.gov/geo/query/acc.cgi?acc=GSE27921>

- Figure Legends (main + EV):

1. Please note that the exact p values are not provided in the legends of figures 3B; EF 4 A-C

Figures 3B: Exact P values are now visualised above the comparisons

Figures 3E: Exact P values of significant changes are now listed in the figure legend

Figures 3F: This is a genome browser view and so no statistics were required

Figure 4A: This is a genome browser view and so no statistics were required

Figure 4B: As there are single data points, no P values can be generated for comparisons

Figure 4C: The previous submission had interactions of specific regions visualised as dot plots, whilst there are differences these were not determined to be significant when data were compared using DESeq2, a more robust measure. As the dot plots were highlighted here, to better represent the differences, subtraction plots have been generated from the Capture-C traces above and the significance indicated in the plots. We hope this is a fairer presentation.

2. 2. Please note that the error bars are not defined in the legends of figures EV4 D, E

Figure EV4D: Error bar statement added

Figure EV4E: Error bar statement added

- Author email bounced

We are not authorised to add personal email addresses to the co-authors in the submission portal. The current institutional emails (the ones that bounced) are not up-to-date and we have no contact with these authors except through their personal emails. We cannot find an updated contact for Samy Alghadban. We have attempted to get his email via different routes but we are still waiting his reply.

-- Samy Alghadban - samy.alghadban@bms.ox.ac.uk (cannot find a contact!)

-- Damien Downes - damien.downes@ndcls.ox.ac.uk damienjdownes@gmail.com

-- Rosa Stolper - rosa.stolper@well.ox.ac.uk rosastolper@hotmail.com

- Sections need to be named and the order should be corrected: Title page - Abstract - Keywords - Introduction - Results - Discussion - Methods - Data Availability - Acknowledgements - Disclosure and Competing Interests Statement - References - Figure Legends - Table(s) - Expanded View Figure Legends.

The order of the sections has been corrected as above

Use the link below to submit your revision:

Referee #1:

The revised paper addressed all the raised concerns.

Referee #2:

The authors sufficiently addressed the comments I raised in their point-by-point response. However, I would prefer to see the reply to two of my major comments incorporated in the manuscript as well, not just in the point-by-point response. In particular, this relates to comments #6 and #7.

In response to comment 7, the authors refer to the unpublished data in the mouse model with the deletion of the Hba-a1/2 genes. Why wasn't this included in the revised manuscript?

The unpublished data of deletion of the Hba-a1/2 genes we refer to in our point-by-point comments is collected from a novel mouse model that has been characterised at further depth than explained in the point-by-point comments. This data will be published soon. We will not refer to this work in this manuscript.

Instead, we are solely referring to published papers which are discussed in response to #7 and which argue for similar conclusions. However, we would like to further highlight the observation in the orthologous human alpha globin cluster, which is already referenced in the text, wherein a deletion spanning the HBA1/2 genes lead to an increase in the expression of a downstream gene (NME4) which could now be interpreted in a new context in the light of the findings presented in this study (Lower et al. 2009).

They performed motif analysis in response to comment 6, why wasn't this included in the manuscript? If these points are addressed, I am happy with the manuscript being accepted for publication.

This analysis has been added as the following sentence found on lines 285-287 and a section has been added to methods to include how the motif analysis was performed.

Referee #3:

I thank the authors for addressing the concerns and revising the manuscript. This work, with the systematic deletions of CTCF sites and the reporter assay elegantly show that

active transcription can serve as an insulator. This is a very valuable insight for the field. The revised text and figures as well as the new analyses and experiments address all my concerns.

I have a few minor, mainly aesthetic, suggestions but otherwise recommend this manuscript for publication.

1) Please check the figure order and references in text:

- Line 159: what does Fig. S1 refer to?

This is referring to Appendix Fig S1, which are visualisations of the CTCF site deletions generated in our mice models.

- Fig 1E is referenced after Fig 2

An earlier reference to Fig 1E has been added on line 173 prior to Fig 2.

- Line 172: There is no figure 1F (did the authors mean 1D?)

Thank you for highlighting this. This is now corrected to 1D.

- EV2B comes after EV3 in text

EV3 is indeed referenced prior to EV2B, however we would like to keep the two panels of EV2A/B together, as it makes it easier for a reader to compare the interaction profiles derived from the Δ HS+44/HS+48 and $\Delta\theta 1/\theta 2$ /HS+44/HS+48 models. As the $\Delta\theta 1/\theta 2$ /HS+44/HS+48 model is introduced later in the text, it is unavoidable to reference EV2B after EV3. I hope the editor would agree this is acceptable.

- Figure 3F is not cited

Fig 3F is cited on line 324.

2) Line 228-230. I think the authors mean to reference Fig 2EVB iii and iv and not Fig 3EVB iii, iv.

Thank you for highlighting this. This is now corrected to Fig 2EVB.

I can now see that comparing the values on the y axis between iii and iv shows that Hba-a1 has a higher interaction frequency with R1 going from ~40% from Hba-a1 (iii) to ~20% in than Hba-a2 (iv). It is not obvious that the values are different for the other enhancers. Can the authors plot the quantification of the values in a bar plot (or another way to plot) for each enhancer to make it easier to compare side by side?

We have added plots showing the differential signals in Fig 2EVB (panels v and vi). These show the subtraction of the signal derived from Hba-a1 and Hba-a2, in both WT (dark grey) and the $\Delta\theta 1/\theta 2$ /HS+44/HS+48 models. This now provides a clearer view of the broader interactions between Hba-a1 and the enhancer region (particularly over R4) – with a positive signal representing regions with more interactions from Hba-a1. This is comparable to the Hba-a1/Hba-a2 specific differential traces generated in prior publication (Davies et al. 2016) as referenced on line 248. Further, when comparing the

differential traces in Fig2EVBv (WT) and the Fig2EVBvi (CTCF Quaddel), the differences are more evidently similar.

3) It is helpful to have Fig EV1 with the TAD structure and all the genes in the locus. If the resolution allows, can the authors also add a zoom-in version for the TAD (region under the black bar in the current EV1) as well? This may also help with the Tiled-C heatmap scaling as now, all the areas outside of the TAD are very dim.

We have added a zoomed view of the locus with ATAC, CTCF ChIP-seq and Tiled-C profiles included under EV1. These zoomed regions correspond to regions plotted in the main figures in the text.

- Davies, J. O., J. M. Telenius, S. J. McGowan, N. A. Roberts, S. Taylor, D. R. Higgs, and J. R. Hughes. 2016. 'Multiplexed analysis of chromosome conformation at vastly improved sensitivity', *Nat Methods*, 13: 74-80.
- Lower, K. M., J. R. Hughes, M. De Gobbi, S. Henderson, V. Viprakasit, C. Fisher, A. Goriely, H. Ayyub, J. Sloane-Stanley, D. Vernimmen, C. Langford, D. Garrick, R. J. Gibbons, and D. R. Higgs. 2009. 'Adventitious changes in long-range gene expression caused by polymorphic structural variation and promoter competition', *Proc Natl Acad Sci U S A*, 106: 21771-6.

Dear Dr. Kassouf,

I am pleased to inform you that your manuscript has been accepted for publication in the EMBO Journal.

You may qualify for financial assistance for your publication charges - either via a Springer Nature fully open access agreement or an EMBO initiative. Check your eligibility: <https://link.springer.com/journal/44318/how-to-publish-with-us>

Yours sincerely,

Cornelius Schneider, PhD
Editor
The EMBO Journal
c.schneider@embojournal.org

Please note that it is The EMBO Journal policy for the transcript of the editorial process (containing referee reports and your response letters) to be published as an online supplement to each paper. If you should prefer removal of any referee-only figures included in the point-by-point response(s), e.g. because they may still be used for future publication or because they have been reproduced from published work by others, please do let us know immediately via response email.

More information is available here: <https://link.springer.com/partners/embo-press/editorial-policies#Peer%20review>